# The augmin complex architecture reveals structural insights into microtubule branching

Erik Zupa [1,3], Martin Würtz [1,3], Annett Neuner[1], Thomas Hoffmann [2], Mandy Rettel [2], Anna Böhler[1], Bram J. A. Vermeulen [1], Sebastian Eustermann [2] ✉, Elmar Schiebel [1] ✉ & Stefan Pfeffer [1] ✉

In mitosis, the augmin complex binds to spindle microtubules to recruit the γ-tubulin ring complex (γ-TuRC), the principal microtubule nucleator, for the formation of branched microtubules. Our understanding of augmin-mediated microtubule branching is hampered by the lack of structural information on the augmin complex. Here, we elucidate the molecular architecture and conformational plasticity of the augmin complex using an integrative structural biology approach. The elongated structure of the augmin complex is characterised by extensive coiled-coil segments and comprises two structural elements with distinct but complementary functions in γ-TuRC and microtubule binding, linked by a flexible hinge. The augmin complex is recruited to microtubules via a composite microtubule binding site comprising a positively charged unordered extension and two calponin homology domains. Our study provides the structural basis for augmin function in branched microtubule formation, decisively fostering our understanding of spindle formation in mitosis.

Microtubules are major components of the eukaryotic cytoskeleton, participating in a variety of biological processes, such as cellular compartmentalisation, chromosome segregation, cell motility, and intracellular transport. Microtubules are assembled from α/β-tubulin dimers that form a 13-protofilament hollow cylinder. In higher eukaryotes, microtubule formation is spatially and temporally controlled by the γ-tubulin ring complex (γ-TuRC). Recent cryo-EM reconstructions have elucidated the structure and molecular architecture of the vertebrate γ-TuRC, revealing a spiral-like overall arrangement of 14 spokes, each containing one copy of γ-tubulin and one of five paralogous gamma-tubulin complex proteins (GCPs)[1–4]. The arrangement of γ-tubulins in the complex approximates microtubule geometry, allowing the γ-TuRC to promote de novo microtubule formation by acting as a structural template during early stages of the nucleation reaction[5].

Microtubule nucleation is focused to distinct subcellular sites, termed microtubule organising centres (MTOCs), most prominently the spindle poles including the centrosome of higher eukaryotes. During mitosis, a dense network of spindle microtubules organises chromosome positioning with respect to the centrosome and subsequently segregates chromosomes between the daughter cells. During this process, microtubules are not only nucleated at the centrosomes, but also from pre-existing microtubules in a process termed microtubule branching[6,7]. Microtubule branching is a complex process requiring the coordinated interplay of several components. As in centrosomal microtubule formation, the nucleation process itself is mediated by the γ-TuRC, but further branching-specific factors are required to promote efficiency of the process and to guarantee correct polarity and direction of branched microtubules in the mitotic spindle. Besides TPX2 (Targeting protein for Xklp2), which was suggested to

[1]Zentrum für Molekulare Biologie der Universität Heidelberg, DKFZ-ZMBH Allianz, Im Neuenheimer Feld 282, 69120 Heidelberg, Germany. [2]European Molecular Biology Laboratory (EMBL), Heidelberg Meyerhofstraße 1, 69117 Heidelberg, Germany. [3]These authors contributed equally: Erik Zupa, Martin Würtz. ✉e-mail: sebastian.eustermann@embl.de; e.schiebel@zmbh.uni-heidelberg.de; s.pfeffer@zmbh.uni-heidelberg.de

increase local α/β-tubulin concentration by condensate formation on the microtubule lattice in *Xenopus laevis*[8,9], the most prominent factor required for microtubule branching is the augmin complex, which was first identified in a genetic screen in *Drosophila* S2 cells[10].

The essential role of augmin in microtubule branching was established across different species in vivo, including plant species[11,12], human cells[13], *Drosophila melanogaster*, where depletion of augmin subunits results in spindle microtubule defects[6,14] and *X. laevis*, where augmin was shown to be central for spindle formation and integrity[7,15]. In human cells, augmin-mediated microtubule branching is not limited to mitosis and meiosis, but it is also centrally involved in organising the non-centrosomal neuronal microtubule networks and in establishing axonal microtubule polarity[16], thereby contributing to development of the central nervous system in mammals[17,18]. Consistently, knock down of the augmin subunit HAUS6 is lethal for mouse embryos[19].

A direct function of augmin as microtubule branching factor could also be demonstrated using purified components from *X. laevis*[20], *D. melanogaster*[21], and human cells[22]. In these experiments, augmin was shown to directly bind to both the microtubule and the γ-TuRC, thus recruiting the γ-TuRC to the pre-existing microtubule and thereby defining both the site of microtubule branching and the orientation of the branched microtubule relative to its carrier[20–22]. Microtubule and γ-TuRC binding capabilities of augmin were attributed to two distinct functional elements, corresponding to two stable hetero-tetrameric subcomplexes of the octameric augmin complex (Fig. 1a): while Tetramer II (TII; containing HAUS2, 6, 7, and 8) mediates microtubule-binding[23,24], Tetramer III (TIII; consisting of HAUS1, 3, 4, and 5) was shown to bind to the γ-TuRC via the adaptor protein NEDD1 (neural precursor cell expressed developmentally down-regulated protein 1)[24,25].

Our understanding of augmin-mediated microtubule branching is strongly hampered by insufficient structural information on the augmin complex. Aiming to fill this gap, we combined cryo-EM and negative stain EM analysis, crosslinking mass spectrometry and neural network-based protein structure prediction to elucidate the molecular architecture and characterise the conformational plasticity of the augmin complex. We identified a composite microtubule binding site with high structural and compositional similarity to the kinetochore-microtubule-binding Ndc80 complex. By acting as an elongated structural scaffold, augmin positions the γ-TuRC on the pre-existing microtubule in a partially flexible manner, allowing the γ-TuRC to nucleate a branched microtubule in an approximate orientation with respect to the mother microtubule.

## Results

### Neural network-based structure prediction generates candidate models for augmin subcomplexes

In order to understand the structural basis for augmin-mediated microtubule branching, we decided to take advantage of the most recent breakthroughs in neural network-based prediction of molecular structures as implemented in AlphaFold[26]. Predicted models for individual HAUS proteins deposited in public databases featured long α-helical segments with no defined folds, indicating that the context of the complex is required for correct prediction of the augmin subunit structures. We therefore decided to predict the structure of the full augmin octamer using AlphaFold-Multimer[27]. However, as judged by the absence of a defined fold for subunits of the TII tetramer and complete lack of interactions between the TII and TIII tetramers, this approach failed (Supplementary Fig. 1).

To reduce the complexity of the system for prediction, we followed a divide-and-conquer approach and decided to break down the augmin complex into two simpler tetrameric subcomplexes. We focused on the augmin TII and TIII tetramers from *X. laevis*, which had been experimentally reported to form stable subcomplexes on their own[24] and are thought to represent two independent functional

modules for microtubule and γ-TuRC binding, respectively[24]. For both tetrameric subcomplexes, all 25 predicted models reached similar predicted local distance difference test (pLDDT) scores (Supplementary Tables 1 and 2) and recapitulated a highly similar overall fold, with only slight variations in global domain arrangement (Fig. 1b–e). Consistently, the per-residue confidence metric showed highly confident prediction for almost all segments of the tetrameric complexes (Supplementary Fig. 2a, b).

### Cryo-EM structural analysis and crosslinking mass spectrometry validate TIII structure prediction

The predicted TIII models featured a highly similar elongated overall structure, characterised by extended coiled-coil segments spanning the full length of the complex (Fig. 1b). Aiming to validate the TIII models predicted by AlphaFold-Multimer using experimental approaches, we purified the *X. laevis* TIII tetramer[24] after recombinant expression (HAUS1-EGFP-HIS$_8$, 2xFLAG-HAUS3, HAUS4 and HAUS5) in insect cells (Fig. 2a and Supplementary Fig. 3). Negative stain EM 2D and 3D class averages (Fig. 2b, c) recapitulated the elongated overall shape predicted by the models and observed previously by negative stain EM[24].

We next subjected the purified augmin TIII tetramer to cryo-EM single particle analysis, obtaining a cryo-EM reconstruction at 7.7 Å global resolution (Fig. 2d, Supplementary Fig. 4, Supplementary Table 3, and Methods section). Strikingly, the predicted TIII models provided an almost ideal fit into the cryo-EM reconstruction, explaining all resolved density segments on secondary structure level (Fig. 2e and Supplementary Table 4). We next optimised the fit of the highest-scoring model into the cryo-EM density by molecular dynamics flexible fitting (MDFF), which resulted in only minor adjustments (Fig. 2f). Notably, several segments of the model were not covered by the cryo-EM density, indicating higher conformational flexibility (Fig. 2e). Consistently, these areas overlapped with the model segments predicted to adopt different conformations by AlphaFold-Multimer (Fig. 1c), indicating that AlphaFold-Multimer may have the capability of identifying regions with increased conformational plasticity.

Subsequently, we sought to further validate our model of the TIII tetramer by crosslinking mass spectrometry[28,29]. After optimising the crosslinking conditions to intermediate crosslinking efficiency with the crosslinking agent BS3 (Supplementary Fig. 5), the sample was subjected to mass spectrometry analysis. We mapped back the 190 highest-scoring crosslinks (intramolecular: 88; intermolecular: 102) to the TIII model after MDFF for evaluation (Fig. 2g, h, Supplementary Fig. 6; and Supplementary Data 1–3; see Methods section). The vast majority of high-confidence crosslinks (91%) was compatible with the AlphaFold-Multimer predictions and cryo-EM structural analysis, further validating the model. Almost all crosslinks violating the distance restraints in the predicted structures were mapped to the model segments observed and predicted to be most flexible (Figs. 1c and 2e, g, and Supplementary Fig. 6).

### Integrated structure and conformational dynamics of the augmin TIII tetramer

Next, we dissected the TIII molecular architecture and conformational dynamics in detail, integrating all available data. The overall elongated structure of the augmin TIII tetramer is characterised almost entirely by extensive coiled-coil segments that are predominantly formed in a pair-wise manner between HAUS3 – HAUS5 and HAUS1 – HAUS4, respectively (Fig. 3a). This is also reflected in the enrichment of high confidence intermolecular crosslinks between the same pairs of HAUS proteins (Fig. 3b) and indicates that HAUS3-HAUS5 and HAUS1-HAUS4 interact more intimately with each other, consistent with previous observations of these pairs of proteins forming stable dimers even in the absence of the other HAUS subunits[24]. HAUS1-HAUS4 and HAUS3-HAUS5, respectively, are aligned from N- to C-terminus almost over

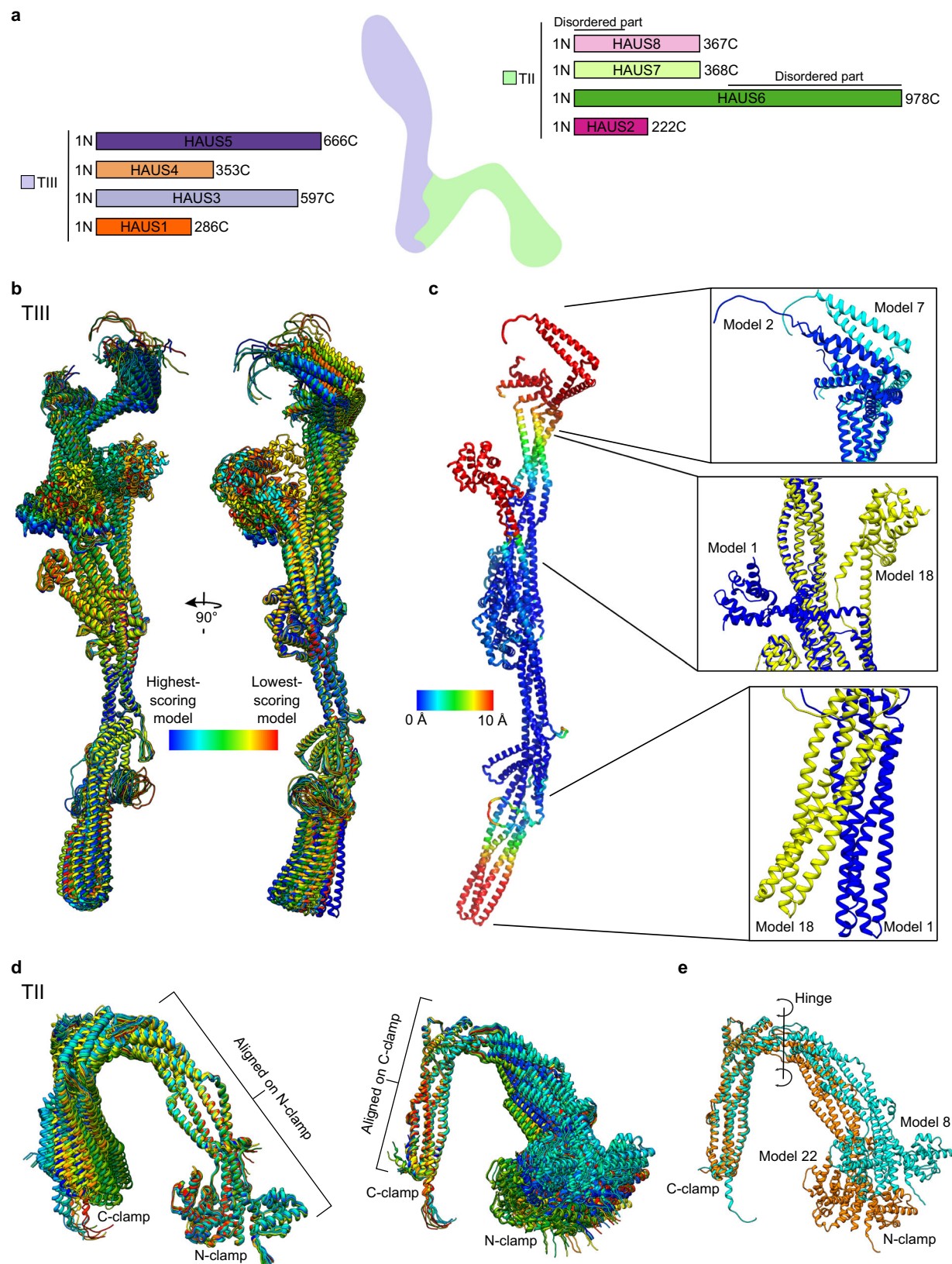

their full length in the complex (Fig. 3a) as also recapitulated in the crosslinking pattern between the two pairs of proteins (Fig. 3b). HAUS3 and HAUS5 fold back onto each other, bringing segments located more towards their protein termini into spatial proximity (Fig. 3a), as confirmed by crosslinks observed between the respective segments of these two proteins (Fig. 3b).

The TIII structure is characterised by a central 'bulge' region, composed of all four HAUS proteins (Fig. 3c), separating the TIII tetramer into two segments. While one of the two segments is assembled exclusively from the HAUS3 and HAUS5 subunits (H3/H5-arm) (Fig. 3d), the other segment is formed by all four HAUS proteins (4H-arm) (Fig. 3e). This arrangement of subunits is also recapitulated in the

**Fig. 1 | Ensemble of models predicted by AlphaFold-Multimer for the tetrameric augmin subcomplexes TIII and TII. a** Domain representation of the HAUS proteins with schematic representation of the augmin holocomplex. TIII and TII tetramers are indicated together with the corresponding HAUS proteins. Disordered regions are annotated. **b** Superposition of predicted models for the augmin TIII subcomplex. The ensemble of models was coloured in rainbow scheme from high (blue) to low (red) prediction score. Colour scheme is given. **c** Left: highest-scoring model coloured according to RMSD against the model with maximum deviation from 0 Å (blue) to 10 Å (red). Colour scheme is given. Right: zoomed views of model segments predicted to be flexible, with the two most extreme conformations shown. **d** Predicted models for the augmin TII subcomplex superposed according to either the N-terminal (left) or C-terminal half (right) of the subcomplex. The ensemble of models was coloured in rainbow scheme from high (blue) to low (red) prediction score. Same colour scheme as in **b**. Disordered regions of HAUS6 (399-978) and HAUS8 (1-154) are not shown. **e** Superposition of TII models with most extreme conformations after alignment according to the C-terminal half of the subcomplex. The apparent hinge between the two halves is indicated (axis with arrows).

crosslinking pattern, in which long segments of HAUS3 and HAUS5 are crosslinked among each other, but not with HAUS1 and HAUS4 (Fig. 3b). The N-terminal segments of all HAUS proteins are located in the bulge region, while their C-terminal segments form the 4H-arm (Fig. 3a). The very C-termini of HAUS1 and 4 are located at the tip of the 4H-arm and are predicted to form a flexible kinked coiled-coil (Figs. 1c and 3e). Consistently, this model segment is not resolved in the cryo-EM reconstruction (Fig. 2e) and high-confidence crosslinks violating the distance restraints are enriched in this area, reflecting the high range of mobility (Fig. 2h and Supplementary Fig. 6). The central segments of HAUS3 and HAUS5 fold into an antiparallel 4-helix coiled-coil bundle that forms the entirety of the H3/H5-arm, the very periphery of which is not resolved in the cryo-EM reconstruction (Fig. 2e), consistent with increased flexibility as suggested by higher root-mean-square deviation (RMSD) in the ensemble of predicted models (Fig. 1c and Supplementary Table 5). The very N-termini of HAUS3 and HAUS5 fold into a small globular α-helical bundle (H3/H5 N-bundle) that is positioned on the surface of the central bulge region in the highest-scoring model (Fig. 3c). In the ensemble of predicted models (Fig. 1b), the H3/H5 N-bundle assumes a continuum of different positions relative to the TIII tetramer core fold. Consistently, the H3/H5 N-bundle is not resolved in the cryo-EM reconstruction (Fig. 2e) and high confidence crosslinks between these segments and residues in a radius of 11 nm can be observed (Fig. 2h and Supplementary Fig. 6).

Cumulatively, integrating structural data from cryo-EM, crosslinking mass spectrometry and neural network-based structure prediction, we could in detail elucidate the structure and conformational plasticity of the augmin TIII tetramer, the functional module for γ-TuRC binding. These data indicate that the core segments of TIII form an overall rigid (Figs. 1c and 2d) structural scaffold likely serving for specific binding and positioning of the γ-TuRC relative to the pre-existing microtubule[24].

## Molecular architecture and conformational dynamics of the octameric augmin holocomplex

Having elucidated the molecular architecture of the isolated TIII tetramer, we analysed in detail the structure predictions for the augmin TII tetramer, the functional module responsible for microtubule binding[23,24]. All predicted models recapitulated a clamp-like overall structure, consisting of two halves separated by a hinge (Fig. 1d, e). The two halves consist of coiled-coil helices formed by the N-terminal (N-clamp) and C-terminal segments (C-clamp) of all HAUS proteins in the TII tetramer, respectively. The N-clamp encompasses a hammerhead-like structure (N-clamp HH) formed by two globular domains of the HAUS6 and HAUS7 N-terminal segments (Fig. 4a).

We next sought to validate the predicted models of the TII tetramer by orthogonal structural data. Since expression of the isolated TII tetramer is reportedly inefficient and structural characterisation by negative stain EM analysis failed[24], we opted for co-expression of all eight augmin subunits to reconstitute the augmin holocomplex. This approach also promised to provide insights into the interaction between TII and TIII tetramers and thereby the complete oct augmin holocomplex architecture. We thus generated an augmin TII construct, with 2xFLAG tag on HAUS6, and used it together with the TIII construct introduced above (including 2xFLAG-HAUS3) for co-expression in

insect cells (Supplementary Fig. 7). SDS-PAGE analysis of the FLAG elution samples indicated that the amount of augmin TIII tetramer dominated over bands of TII tetramer proteins. Using size exclusion chromatography (SEC), we could separate the augmin octamer from the augmin TIII tetramer subcomplex (Supplementary Fig. 7c, d).

Negative stain EM 2D class averages of the augmin holocomplex (Fig. 4b and Supplementary Fig. 7e) recapitulated the overall elongated shape of the TIII tetramer, and additional density for the TII tetramer could be clearly discerned, overall reflecting the h-like shape of the augmin octam holocomplex described before[23,24]. We subjected the negatively stained augmin particles to 3D classification to obtain a 3D reconstruction of the augmin holocomplex in a defined conformational state, in which the TIII tetramer could be unambiguously identified (Fig. 4c). The remaining density clearly reflected the clamp-like overall shape of the predicted TII tetramer models and the hammerhead-like region formed by the N-termini of TII HAUS proteins was recapitulated in the density, allowing us to confidently determine the orientation of TII in the 3D reconstruction (Fig. 4c). The approximate placement of the predicted TII tetramer into the negative stain 3D class indicated an interaction between the TIII H3/H5-arm and the TII C-clamp.

Guided by these insights into the overall organisation of the augmin holocomplex, we used AlphaFold-Multimer to predict the structure of the TIII H3/H5-arm in combination with the TII tetramer (Fig. 5a and Supplementary Table 6). In all predicted models, the TII C-clamp interacts with the periphery of the TIII H3/H5-arm in a manner that is fully compatible with the negative stain 3D reconstruction of the augmin holocomplex, as judged by seamless fit of the composite TIII tetramer-TII C-clamp model into the EM density (Fig. 5b), thus providing an accurate model for the interface between the TII and TIII tetramers.

To complete structural characterisation of the augmin holocomplex, we next focused on the TII N-clamp. Notably, in none of the predicted models, the TII N-clamp orientation was compatible with the EM density (Supplementary Fig. 8), indicating that AlphaFold-Multimer was not capable of correctly predicting the relative arrangement of the TII N- and C-clamp around the hinge. We thus individually docked the TII N-clamp as a rigid body into the remaining density segments guided by the orientation of the N-clamp HH (Fig. 5b). Merging the TII N-clamp with the model for the TII C-clamp in complex with TIII, we generated a model for the complete augmin holocomplex (Fig. 5c).

The N-clamp is linked to the TII C-clamp by a hinge region, which allows for an opening and closing motion of the TII clamp, as suggested by the ensemble of predicted models (Figs. 1d and 5a). Consistently, negative stain 2D classes of the augmin holocomplex reflected a range of TII conformations (Fig. 4b and Supplementary Fig. 7e) and 3D classification provided EM densities representing the TII clamp in at least two different conformational states (Supplementary Fig. 9a), with a ratio of 1:1 as judged by number of particles contributing to the two individual classes (Methods section). Following the same modelling approach as described above, we also generated a model for the second TII conformation (Supplementary Fig. 9b–d). The two experimentally observed TII conformations are related by an opening/closing of the TII clamp of approximately 23° (Fig. 5d).

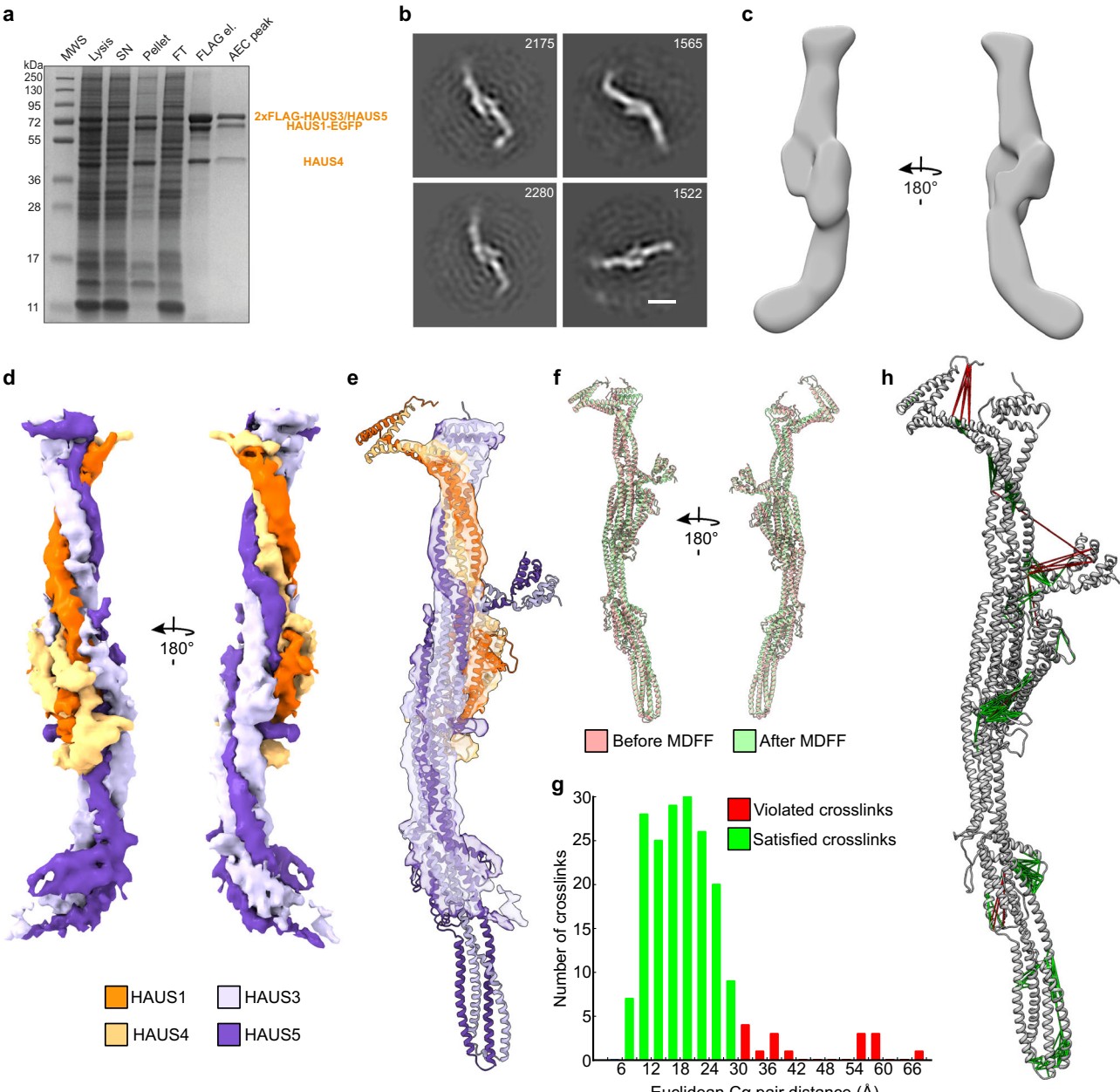

**Fig. 2 | Cryo-EM structural analysis and crosslinking mass spectrometry validate the predicted TIII models. a** SDS-PAGE analysis of the purified augmin TIII subcomplex. MWS: molecular weight standard; Lysis: Cell lysate; SN: supernatant; Pellet: cell pellet; FT: flow-through after incubation with FLAG beads, FLAG el.: FLAG elution; AEC peak: peak fraction after anion exchange chromatography (Supplementary Fig. 3). Purified proteins are indicated (orange). Augmin TIII tetramer purification and SDS-PAGE analysis were repeated at least twice with similar results (*n* = 3 experiments). **b**, **c** Negative stain-EM analysis of augmin TIII. **b** Selected 2D class averages, particle numbers are given, scale bar 10 nm. **c** 3D reconstruction of

the augmin TIII subcomplex. **d** Segmented cryo-EM density of the augmin TIII tetramer. Colouring scheme is given. **e** Fit of the predicted model after MDFF into the density. Same colouring as in **c**. **f** Superimposition of the highest-scoring predicted model before and after MDFF. **g** The 190 highest-scoring crosslinks were mapped back to the TIII structure and the distance between crosslinked residues was plotted. Satisfied (green) and violated (red) crosslinks according to a distance threshold of 30 Å. **h** Visualisation of the same set of crosslinks mapped back to the augmin TIII model with same colour scheme as in **g**. Source data are provided as a Source Data file.

## The TII N-clamp is a composite microtubule-binding site

Finally, we aimed to understand how TII interacts with the pre-existing microtubule to anchor the γ-TuRC for nucleation of spindle microtubules. Several studies have established that the disordered, positively charged HAUS8 N-terminus (141 residues) is essential for the augmin-microtubule interaction[23,24,30], as demonstrated by complete inability of recombinant augmin variants lacking the HAUS8 N-terminus to bind to microtubules[23]. However, the microtubule binding affinity of the HAUS8 N-terminus alone is approximately an order of magnitude weaker as compared to the HAUS6-HAUS8 dimer

or the complete TII tetramer[23], indicating that additional contacts between TII and the microtubule lattice are important for stable augmin recruitment.

To identify these additional contact areas in TII, we analysed the structure of the N-clamp HH, from which the disordered HAUS8 N-terminus projects, in more detail. The N-clamp HH is formed by the N-terminal domains of HAUS6 and HAUS7, which were predicted to fold into highly similar globular domains that have high structural similarity to the calponin homology (CH) domain of the kinetochore-microtubule binding protein Ndc80 (Supplementary Fig. 10a). CH

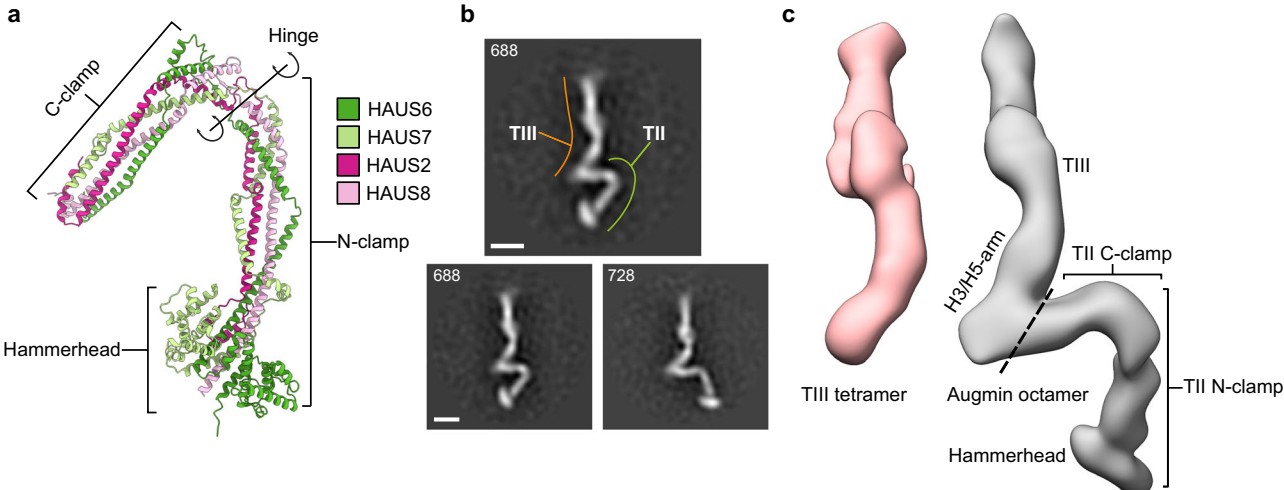

**Fig. 3 | Structure of the augmin TIII tetramer. a** Left: schematic representation of the subunit organisation of the augmin TIII tetramer. 4H-arm, bulge region, H3/H5-arm and the N- and C-termini of HAUS proteins are indicated. Right: model of the augmin TIII tetramer after MDFF. **b** Bar plot representation of crosslinks from Fig. 2g between individual pairs of HAUS proteins. **c**–**e** Zoomed views focusing on individual regions of the augmin TIII tetramer, i.e. the bulge region (**c**), the H3/H5 arm (**d**) and the 4H-arm (**e**). Colour scheme as in **e**.

**Fig. 4 | TII binds to the TIII H3/H5-arm. a** Highest-scoring predicted model of the augmin TII tetramer. Structural features are indicated and colour scheme is given. **b** Representative negative stain EM 2D class averages of the augmin octamer showing two distinct conformations of the TII-tetramer. Scale bar 10 nm, particle numbers are given, TIII tetramer (orange) and TII tetramer (green) are indicated. **c** Negative stain 3D reconstructions of the isolated TIII tetramer (red) and the augmin holocomplex in one of two main conformations (grey). Structural features are annotated and the boundary between TII and TIII in the augmin holocomplex is indicated (dashed line).

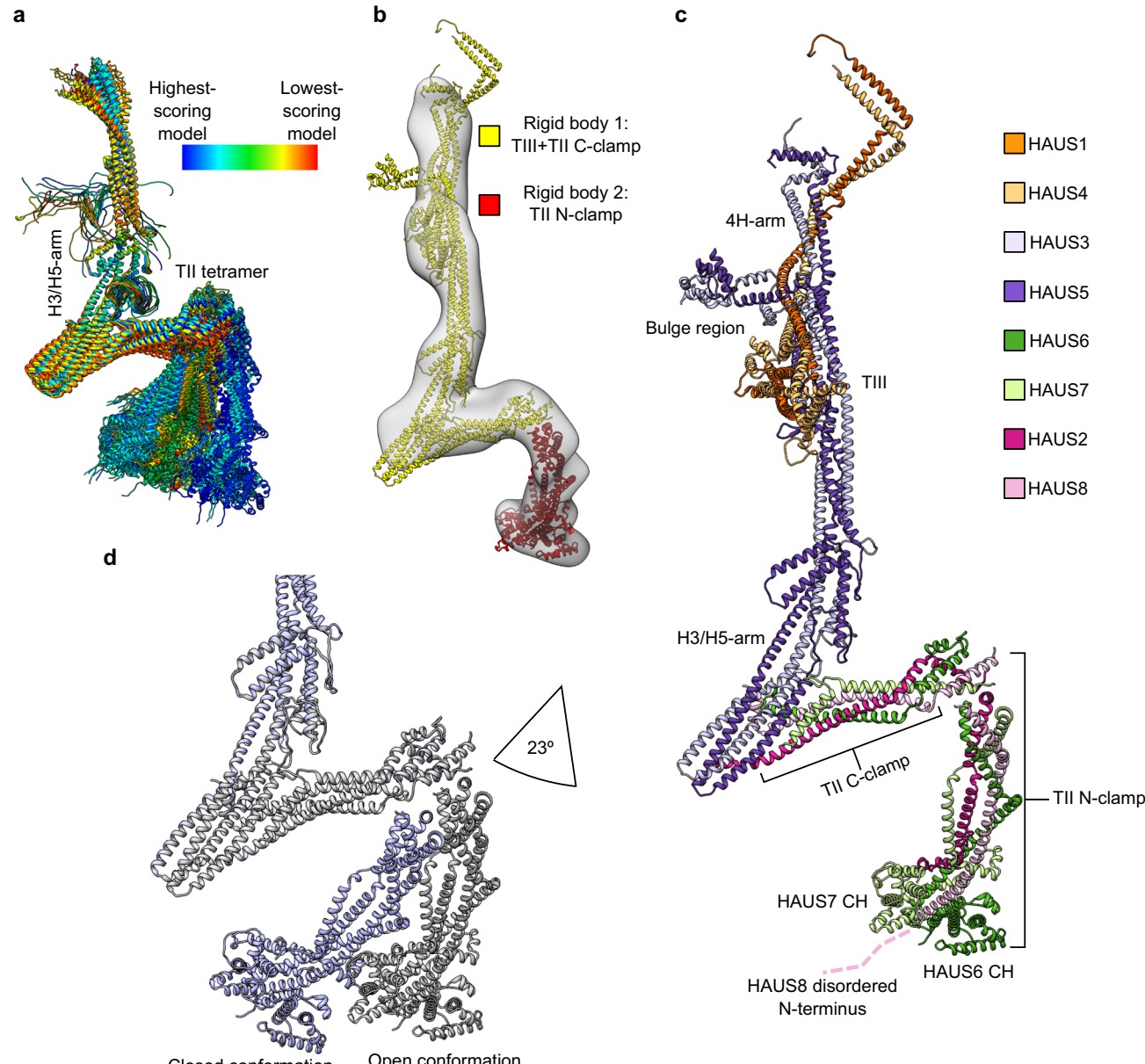

**Fig. 5 | Subunit architecture and conformational plasticity of the augmin holocomplex.** **a** Superposition of predicted models for the augmin TIII H3/H5 arm in complex with TII. The ensemble of models was coloured in rainbow scheme from high (blue) to low (red) prediction score. Colour scheme is given. **b** Two model segments were fitted into the negative stain EM 3D reconstruction of the augmin holocomplex as rigid bodies: TIII + TII C-clamp (yellow); TII N-clamp (red). **c** Subunit architecture of the complete augmin holocomplex in the open conformation. Structural features are indicated. Colour scheme is given. **d** The augmin holocomplex structures in open (grey; from panel **c**) and closed conformations (purple; from Supplementary Fig. 9), superposed according to TIII. Only TII and the TIII H3/H5-arm are shown. The two conformations are related by opening/closing of the TII clamp by 23°.

domains have been characterised as microtubule binding domains in different microtubule binding proteins, including EB1 (end binding 1), EB3[31,32], Ndc80[33], and Nuf2[34]. The CH domains of these proteins share 10–20% of sequence identity among each other and with the HAUS6 and HAUS7 CH domains (Supplementary Table 7). This indicates that the CH domains of HAUS6 and HAUS7 likely underlie the microtubule binding function of the TII tetramer, in concert with the unordered positively charged N-terminus of HAUS8. Such a binding mode rationalises the strong synergy in microtubule binding between the HAUS8 N-terminus and the other TII subunits from a structural perspective[23].

## Discussion
By combining neural network-based structure prediction, negative stain EM, cryo-EM, and crosslinking mass spectrometry, we elucidated

the molecular architecture and defined the conformational plasticity of the augmin complex, providing structural insights into augmin-mediated microtubule branching.

The combination of a broad set of structural biology-related methods proved to be crucial for elucidating the augmin structure. While the cryo-EM reconstruction provided a conceptual understanding of how the elongated overall structure of the TIII tetramer is established by extensive coiled-coil segments, resolution of the reconstruction was clearly not sufficient for detailed dissection of the subunit architecture and de novo assignment of the HAUS proteins. Similarly, while the crosslinking mass spectrometry data provided a detailed network of intra- and inter-subunit proximities, the highly intertwined structure of the augmin TIII tetramer posed a very challenging case and we could not elucidate the subunit architecture in

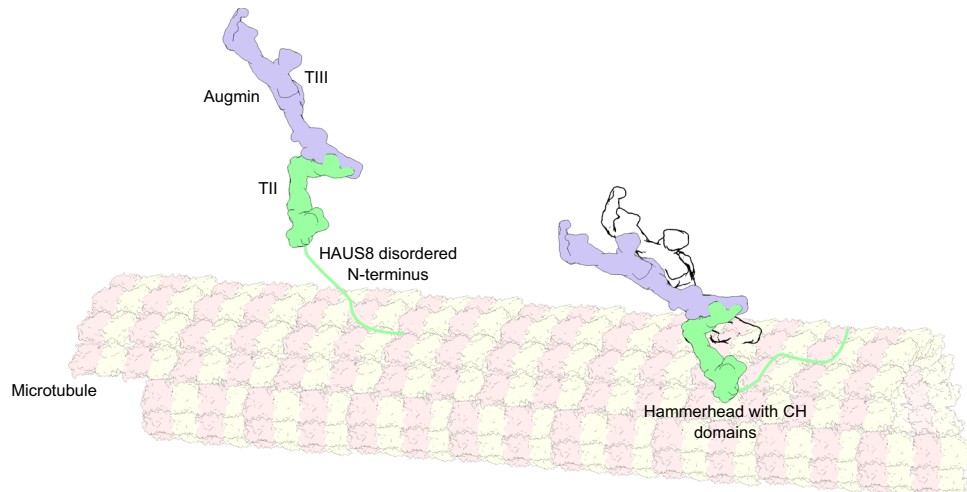

**Fig. 6 | Structure-guided hypothetical model for augmin recruitment to the microtubule.** Left, the augmin holocomplex is recruited to the pre-existing microtubule via the positively charged and flexible HAUS8 N-terminus, which is essential for microtubule binding[23,30]. Right, the augmin holocomplex is stabilised and oriented on the microtubule lattice via HAUS6/HAUS7 CH domain binding, as supported by high synergy of the HAUS8 N-terminus and the remaining TII segments in microtubule binding[23]. Conformational flexibility of the TII clamp (indicated by the black outline) may allow to flexibly adjust the nucleation angle of the branched microtubule.

detail based on these proximities, even when interpreting them in the context of the cryo-EM reconstruction. While AlphaFold-Multimer[27] predicted surprisingly accurate models for the two individual augmin subcomplexes TII and TIII, it failed to predict the structure of the octameric augmin holocomplex and the relative arrangement of the TII N- and C-clamp segments around the hinge region correctly. Thus, orthogonal structural data are still required to address the structure of larger oligomers and to validate the predicted structures[35,36]. To this end, while our negative stain EM densities of the augmin holocomplex allowed for reliable characterization of the overall augmin architecture, follow-up studies will be required in particular to understand the structure and conformational dynamics of the TII hinge region in more detail. Surprisingly, even though not designed for this purpose, AlphaFold-Multimer was able to successfully identify protein segments with increased conformational plasticity, indicating that future developments of protein structure prediction may have the potential to provide information on protein dynamics.

The structure of the augmin complex comprises two structural elements with distinct but complementary functions during microtubule branching. On the one hand, the TIII tetramer in combination with the TII C-clamp forms a predominantly rigid structural scaffold large enough to encompass and position the ~25-nm-sized γ-TuRC at a defined distance and an approximate orientation with respect to the pre-existing microtubule. On the other hand, the TII N-clamp anchors this γ-TuRC-binding element on the pre-existing microtubule via HAUS8 and the HAUS6/7 CH domains. The TII hinge region separating the two clamp elements may allow for a defined range of flexibility in the relative positioning of the γ-TuRC with respect to the pre-existing microtubule. Using TIRF microscopy to analyze the geometry of branched microtubule networks in vitro[20,22] and also in vivo[37] it has been established that the inter-microtubule angles after branching have a spread of approximately 20-30° around the mean branching angle. This is in good agreement with the opening angle experimentally observed (23°; Fig. 5d) for the TII clamp, suggesting that plasticity of the TII hinge region may directly define the variability of microtubule branching angles.

While our structural understanding of the interaction between the augmin complex and the γ-TuRC is very limited, the identification of microtubule-binding CH domains in the HAUS6 and HAUS7 N-termini as part of a putative composite microtubule binding site in the TII N-clamp provides a framework for approximate positioning of the

augmin complex on the pre-existing microtubule. Comparing the predicted structure of CH domains in HAUS6 and HAUS7 with all available structures of microtubule-binding CH domains (Ndc80, EB1, EB3), similarity to Ndc80 is most apparent (Supplementary Fig. 10). The CH domain of Ndc80 has been structurally characterised in complex with the microtubule lattice by cryo-EM, revealing a repetitive binding pattern along the distal side of single protofilaments, at the interfaces between individual α- and β-tubulin subunits (Supplementary Fig. 11a)[38–40]. To extrapolate a structural model for augmin binding to the microtubule lattice, we superposed the HAUS6/7 CH domains to the Ndc80 CH domains on the microtubule lattice. Intriguingly, we observed that only one of the two HAUS CH domains can bind to the microtubule lattice at the same time in the conformation predicted for the N-clamp HH (Supplementary Fig. 11b). However, rotation of the two HAUS CH domains around their centroids by ~180° relative to each other would transition the CH domains into a configuration compatible with binding of the CH domains in tandem as experimentally observed for Ndc80 (Supplementary Fig. 11c). Whether indeed both CH domains are engaged in microtubule binding, in which configuration they are positioned with respect to microtubule polarity and whether the CH domain rearrangements required for simultaneous binding may represent a real conformational change during microtubule binding or rather result from inaccurate CH domain orientation during AlphaFold-Multimer prediction, remains to be investigated. Experimental structures of the augmin complex bound to microtubules will provide insights into these open questions. Of note, the microtubule binding mode of EB3 has also been elucidated by cryo-EM, revealing CH domain binding into the groove between neighbouring protofilaments (Supplementary Fig. 11a)[41], but much higher structural similarity between the HAUS6/7 CH domains and the Ndc80 CH domain argues against such a binding mode for augmin.

Not only the presence of two CH domains but the entire molecular organisation of the composite microtubule binding site in augmin is highly reminiscent of the kinetochore complex, where two CH domains contributed by Ndc80 and Nuf2 (corresponding to the HAUS6 and HAUS7 CH domains) and an unordered positively charged extension contributed by Ndc80 (corresponding to the HAUS8 N-terminus) are required for efficient microtubule-binding. Moreover, the microtubule binding affinity of both Ndc80 and augmin is fine-tuned by a common mechanism, namely by phosphorylation of the unstructured, positively charged microtubule binding domains of

Ndc80[42,43] and HAUS8[44]. Interestingly, it was shown that components of the augmin and Ndc80 complexes can interact[45,46]. As augmin-mediated microtubule branching was shown to be important for kinetochore fibre formation and integrity[6,47–49], it will be interesting to further investigate if there is a functional interplay of these two microtubule binding complexes with highly similar microtubule binding capabilities.

Finally, we integrated our structural data with available interaction data to propose a hypothetical model for augmin recruitment to microtubules (Fig. 6). TII binding to microtubules was demonstrated to be fully dependent on the presence of the HAUS8 N-terminus, although the HAUS8 N-terminus is binding to microtubules with comparably low affinity[23], which together may suggest a role of the highly flexible HAUS8 N-terminus in establishing the initial interactions between augmin and the microtubule. Binding affinity of the TII tetramer to microtubules is 10 times higher as compared to the isolated HAUS8 N-terminus[23], which clearly indicates that additional TII components are required for stable binding. These were identified in our study to most likely correspond to the CH domains of HAUS6 and HAUS7, located in the N-clamp HH. Thus, initial augmin binding via the HAUS8 N-terminus may potentially be followed by stabilisation of the interaction and positioning of augmin on the microtubule for the branching reaction by HAUS6/7 CH domain binding. Such a binding process involving two different modes of interaction would be consistent with the dynamics of augmin binding to microtubules as recently observed in vitro using TIRF microscopy, where augmin has the capability to diffuse along the microtubule before it stably binds and the branching reaction occurs[22].

Overall, our study provides structural insights into the augmin molecular architecture, which can serve as reliable framework for follow-up studies that will bring the γ-TuRC in context and thus ultimately lead to mechanistic understanding of microtubule branching.

## Methods
### DNA cloning
Cloning of the MultiBac[TM] (GENEVA Biotech) constructs, all with polH expression cassettes, was done as described previously[50,51], tagging individual proteins inspired by[24]. Genes of the *X. laevis* augmin TII and TIII complexes (HAUS1 (XM_018267162.1), HAUS2 (NP_001085195.1), HAUS3 (XM_018226568.1), HAUS4 (NM_001096090.1), HAUS5 (XM_018226568.1), HAUS6 (NM_001097095.1), and HAUS7 (NP_001121229.1)) were purchased (Integrated DNA Technologies IDT, USA) optimised for insect cell expression. cDNA of HAUS8 (XB-GENEPAGE-579) was a kind gift of Simone Reber (IRI, Humboldt University Berlin). To facilitate cloning, each gene was ordered with specific overhangs: 5′ AAAACCTATAAAT and 3′TCTAGAGCCTGCAGT (HAUS4, HAUS5); 5′ AAAACCTATAAATATGGACTACAAGGACGATGACGACAAGGATTACAAGGATGACGACGATAAGATCCCAACGACCGAAAACCTGTATTTTCAGGGCGCCATG and 3′TCTAGAGCCTGCAGT (HAUS3 and HAUS6, additional N-terminal 2xFLAG-TEV tag); 5′ATTCCCCTCTAGAAATA and 3′GAAAACCTGTATTTTCAGGGCGGATCCGCTGGCTCCGCT (HAUS1, for cloning into pet26b-EGFP vector). Ordered genes were cloned without prior PCR amplification into MultiBac[TM] vectors via NEB-builder reaction (NEB) following the standard protocol. All PCR reactions were done using Q5 DNA polymerase (NEB) following the standard protocol. MultiBac[TM] vectors and HAUS8 were amplified with primers listed in Supplementary Table 8. To introduce a C-terminal TEV-EGFP-3C-8xHis tag on HAUS1, the ordered gene was directly cloned into pet26b-EGFP (Supplementary Table 9 and then recloned into pIDS[polH] vector using primers listed in Supplementary Table 8. For tetramer TIII, 2xFLAG HAUS3, HAUS4 were cloned into pIDK[polH] and HAUS5 into pACEBac1. The expression cassettes of 2xFLAG HAUS3 and HAUS4 in pIDK[polH] were combined via NEB-builder reaction (NEB) using the combination primers listed in Supplementary Table 8. Afterwards, the plasmids were combined into a single construct using Cre-recombination (Cre-recombinase, NEB) following the MultiBac[TM] manual (GENEVA Biotech, version 5.1), resulting in construct augmin-TIII: pACEBac1, HAUS5; pIDK[polH], 2xFLAG HAUS3, HAUS4; pIDS[polH], HAUS1-EGFP-8xHis. Similarly, the augmin TII construct consists of Cre-recombined: pACEBac1, HAUS7, 2xFLAG-HAUS6; pIDS[polH], HAUS2, HAUS8. All constructs were verified via PCR amplification and sequencing. The two constructs were used for bacmid production in DH10EmBacY cells following manufacturer instructions (GENEVA Biotech version 5.1).

### Protein expression
Baculoviruses were produced in Sf21 insect cells (EMBL Protein Expression and Purification Core Facility, Heidelberg Germany) using cellfectin II reagent (Thermo Fisher Scientific). The second virus generation was amplified in 50 ml ($1 \times 10^6$ cells/ml) culture. Afterwards, baculoviruses were harvested and diluted 1:100 in 100–400 ml ($1–2 \times 10^6$ cells/ml) expression culture in Sf21 or High Five cell line (BTI-TN-5B1-4, cat. no. B855-02, Invitrogen) using Sf-900 III medium (Thermo Fisher Scientific) supplemented with 100 units/ml penicillin/100 µg/ml streptomycin (Thermo Fisher Scientific). Expression was done shaking at 27 °C for 60 h. Cells were harvested via centrifugation (800×$g$ for 5 min), flash frozen in liquid $N_2$ and stored at −80 °C until further usage.

### Protein purification
For protein purification, cells were resuspended in cold lysis buffer, which was different for EM experiments (20 mM Tris-HCl pH 7.5, 150 mM NaCl, 1 mM $MgCl_2$, 1 mM EGTA, 1 mM DTT, 0.05% (v/v) Tween 20) than for cross-linking MS (20 mM HEPES pH 7.4, 150 mM NaCl, 4 mM $MgCl_2$, 1 mM EGTA, 0.5 mM DTT); in both cases, it contained one cOmplete EDTA-free protease inhibitor tablet (Merck) per 15 ml lysis buffer. Resuspended cells were sonicated ($3 \times 1$ min with 0.6 amplitude, Hielscher UP50H) and centrifuged at 20,000×$g$ for 30 min at 4 °C. The supernatant was filtered (Whatman sterile filters 0.45 µm pore size) and incubated for 2 h rotating at 4 °C with Anti-FLAG M2 Affinity Gel (Sigma-Aldrich). Anti-FLAG beads were separated from the flow-through via centrifugation (800×$g$, 3 min at 4 °C) and washed one time with lysis buffer and two times with wash buffer (EM experiments: 20 mM Tris-HCl pH 7.5, 150 mM NaCl, 1 mM $MgCl_2$, 1 mM EGTA, 0.5 mM DTT; cross-linking MS experiments: 20 mM HEPES pH 7.4, 150 mM NaCl, 4 mM $MgCl_2$, 1 mM EGTA, 0.5 mM DTT). Protein complexes were eluted two times from FLAG beads by incubating for 20–30 min with wash buffer supplemented with 0.5 mg/ml 3xFLAG peptide (Gentaur) followed by centrifugation (800×$g$, 3 min at 4 °C). Elutions were pooled and used for SDS-PAGE analysis (4–20% gradient or 10% SDS Precast-gel (BIO-RAD) stained with Coomassie Brilliant Blue G250 (Sigma Aldrich)) and subsequent purifications steps. SDS page gel images were acquired using LAS4000IR (software version v2.1) and analysed using Fiji[52] (ImageJ v2.1.0/1.53c).

### Anion-exchange chromatography
FLAG elutions of augmin TIII were loaded onto an anion exchange column (Mono Q® 5/50 GL or Capto[TM] HiRes Q 5/50, Cytiva) equilibrated either with buffer (EM experiments: 20 mM TRIS-HCl pH 7.5, 150 mM NaCl, 1 mM $MgCl_2$, 1 mM EGTA, 0.5 mM DTT; cross-linking MS experiments: 20 mM HEPES pH 7.4, 150 mM NaCl, 4 mM $MgCl_2$, 1 mM EGTA, 0.5 mM DTT). Afterwards, complexes were eluted with a flow rate of 0.5 ml/min with a gradient from 150 mM NaCl to 1 M NaCl concentration over 20 column volumes. The peak fractions were verified via SDS-PAGE, afterwards combined and either concentrated (Amicon 30 K), aliquoted and used for further experiments or flash frozen in liquid $N_2$ and stored at −80 °C. Runs were performed using an ÄktaGo instrument (Cytiva) controlled by Unicorn software (version 7.6) and data were processed using Microsoft Excel (v16.46.21021202) and plotted using Prism software (version 9.1).

## Size exclusion chromatography

FLAG elutions of the augmin octamer sample were run on a size exclusion column (Superose6 increase 10/300 GL, Cytiva) equilibrated with buffer (EM experiments: 20 mM TRIS·HCl pH 7.5, 150 mM NaCl, 0.5 mM DTT; cross-linking MS experiments: 20 mM HEPES pH 7.4, 150 mM NaCl, 4 mM MgCl$_2$, 1 mM EGTA, 0.5 mM DTT). Run was controlled via Unicorn software (version 7.6) on ÄktaGo instrument (Cytiva) at a constant flow rate of 0.25 ml/min. In independent runs of the SEC column, blue dextran 2000 (Cytiva) was used to determine the void volume (8.8 ml) and thyroglobulin 669 kDa (13.2 ml) and aldolase 158 kDa (16.3 ml) as size markers. The peak fractions were verified via SDS-PAGE, afterwards combined, concentrated (Amicon 30 K) and used for further experiments or flash frozen in liquid N$_2$ and stored at −80 °C. Data were processed using Microsoft Excel (v16.46.21021202) and plotted using Prism software (version 9.1, GraphPad).

## Negative stain EM

5 µl of sample were applied on glow-discharged copper-palladium 400 EM mesh grids covered with an approximately 10 nm thick continuous carbon layer (G2400D, Plano GmbH). After 30 s incubation at room temperature, grids were blotted with a Whatman filter paper 50 (CAT N.1450-070) and washed with three drops of water. Sample on grids was stained with 3% uranyl acetate in water. Images were acquired on a Talos L120C TEM equipped with a 4k × 4 K Ceta CMOS camera (Thermo Fisher Scientific). Data were acquired using EPU (v2.9, Thermo Fisher Scientific) at a nominal defocus of about −2 µm and an object pixel size of 0.328 nm or 0.2552 nm.

Image processing was performed in Relion 3.1[53] for all datasets. The contrast transfer function (CTF) of micrographs was estimated using Gctf[54]. Afterwards, approximately 500 particles were selected manually to create an initial 2D class for automated particle picking. After automated particle picking 2D classification was performed in 20−200 classes, using a T-factor of 2, and a translational search range of 20 pixels at 2 pixels increment and a mask of 400−650 Å diameter.

For the augmin TIII data after AEC, 504 images were acquired. 412,188 particles were picked automatically and extracted at 6.56 Å pixel size. Extracted particles were sorted by 4 subsequent rounds of 2D classification, where always the best true positive classes were selected. Afterwards, 43,794 particles were re-extracted at 3.28 Å pixel size and subjected to another round of 2D classification. From this, 13,594 particles were selected for 3D classification.

For the augmin octamer dataset (583 images), 80,837 particles were picked automatically and subjected to two consecutive runs of 2D classification at full spatial resolution (0.2552 nm). In total, 56,021 particles were selected and used for a final round of 2D classification as well as for 3D classification.

The 3D classification runs were performed using the cryo-EM density of the augmin TIII tetramer low-pass filtered to 60 Å resolution as initial reference, with a mask diameter of 450 Å (TIII) or 600 Å (octamer), a T-factor of 4, 6 classes and an offset search range of 20 pixels with an offset search step of 2 pixels. For the TIII data set 11,897 particles were selected after initial 3D classification. For the octamer dataset, two individual classes were selected containing 21% (11,969) and 19% (10,658) of particles, respectively. The final sets of particles were subjected to individual 3D refinement runs, post-processing and one additional round of 2D classification without image alignment to characterise conformational plasticity of the augmin octamer in 2D. The pixel size of the negative stain EM 3D reconstructions was changed from nominal 2.552 Å to calibrated 2.35 Å prior to docking of atomic models.

## Cross-linking MS

50 µg of AEC-purified TIII tetramer was crosslinked for 15 min (RT, 600 rpm) with 0.1 mM bis[sulfosuccinimidyl] (BS3, Thermo Fisher). Reaction was stopped by adding Tris-HCl to a final concentration of 100 mM. Afterwards, sample was incubated for 30 min (50 °C, 600 rpm) with dithiothreitol (DTT) concentration adjusted to 10 mM. Then, 2-chloroacetamide (CAA) was added to a final concentration of 50 mM and reaction was incubated for 30 min at RT protected from light. Subsequently, trypsin (Promega, V511A) was added (1:50, trypsin to protein concentration) and incubated for 4 h at 37 °C and reaction was stopped by adding trifluoroacetic acid (TFA) to a final concentration of 1% (v/v). Digested peptides were concentrated and desalted using an OASIS® HLB µElution Plate (Waters) according to manufacturer instructions. Crosslinked peptides were enriched using size exclusion chromatography[55]. Briefly, desalted peptides were reconstituted with SEC buffer (30% (v/v) acetonitrile (ACN) in 0.1% (v/v) TFA) and fractionated using a Superdex Peptide PC 3.2/30 column (Cytiva) on a 1200 Infinity HPLC system (Agilent) at a flow rate of 50 ml/min. Fractions eluting between 50-70 ul were evaporated to dryness and reconstituted in 30 µl 4% (v/v) ACN in 1% (v/v) formic acid (FA).

Collected fractions were analysed by liquid chromatography (LC)-coupled tandem mass spectrometry (MS/MS) using an UltiMate 3000 RSLC nano LC system (Dionex) fitted with a trapping cartridge (µ-Precolumn C18 PepMap 100, 5 µm, 300 µm i.d. x 5 mm, 100 Å) and an analytical column (nanoEase™ M/Z HSS T3 column 75 µm x 250 mm C18, 1.8 µm, 100 Å, Waters). Trapping was carried out with a constant flow of trapping solvent (0.05% TFA in water) at 30 µL/min onto the trapping column for 6 min. Subsequently, peptides were eluted and separated on the analytical column using a gradient composed of Solvent A ((3% DMSO, 0.1% formic acid in water) and solvent B (3% DMSO, 0.1% formic acid in acetonitrile) with a constant flow of 0.3 µL/min. The outlet of the analytical column was coupled directly to an Orbitrap Fusion Lumos (Thermo Scientific) mass spectrometer using the nanoFlex source. The peptides were introduced into the Orbitrap Fusion Lumos via a Pico-Tip Emitter 360 µm OD x 20 µm ID; 10 µm tip (CoAnn Technologies) and an applied spray voltage of 2.1 kV, instrument was operated in positive mode. The capillary temperature was set at 275 °C. Only charge states of 4-8 were included. The dynamic exclusion was set to 30 s. and the intensity threshold was 5e$^4$. Full mass scans were acquired for a mass range 350−1300 m/z in profile mode in the orbitrap with resolution of 120,000. The AGC target was set to Standard and the injection time mode was set to Auto. The instrument was operated in data dependent acquisition (DDA) mode with a cycle time of 3 s between master scans and MS/MS scans were acquired in the Orbitrap with a resolution of 30,000, with a fill time of up to 100 ms and a limitation of 2e5 ions (AGC target). A normalised collision energy of 32 was applied. MS2 data was acquired in profile mode.

## Cross-linking MS data analysis

All data were analysed using the cross-linking module in Mass Spec Studio v2.4.0.3524[56]. Parameters were set as follows: Trypsin (K/R only), charge states 3−7, peptide length 6−50, percent Evalue threshold = 70, MS mass tolerance = 10 ppm, MS/MS mass tolerance = 10, elution width = 0.5 min. BS3 cross-link residue pairs were constrained to K on one end (to reduce search space) and one of KSTY on the other. N-termini of proteins were not considered for crosslinks. Higher energy collisional dissociation (HCD) fragmentation was used for MS acquisition and multiple identifications for a given pair of peptide sequences was permitted only if chromatographic separation from any other identification exceeded one chromatographic peak width (i.e., >0.3 min). Retained peptide spectra matches were manually validated according to MS spectra quality, MS/MS spectra quality, mass accuracy, peak assignment in MS/MS spectra and fragmentation of cross-link product (ions coverage of crosslinked peptides). At this step, intra- and inter-molecular crosslink candidates were separated for FDR estimation. Only cross-links with a q-value corresponding to <5% FDR at the peptide pair and residue pair level were retained. Afterwards, the data were exported from Mass Spec Studio to obtain a list of crosslinks with a unique pair of potential residue sites (Supplementary Data 1).

The exported crosslinks were ranked according to an aggregate score based on the sum of spectrum scores that makeup the aggregate item. Spectrum scores are OMSSA[57]-based and reported as the -ln(Expectation). In case of multiple identifications in the different replicates, only the copy of each crosslink site with higher aggregate score was extracted from the exported file (Supplementary Data 2). Crosslink sites with an aggregate score higher than 250 (Supplementary Data 3) were used for model validation and preparation of bar plots. Bar plots were prepared in Xvis[58] and model validation was performed using Xlink Analyzer[59] setting a distance threshold to 30 Å. Scores for validation using Xlink Analyzer were normalised by a factor of 10.

## Cryo-EM data acquisition for augmin TIII tetramer

4 μl of purified augmin TIII tetramer were applied on Cu R2/1 holey carbon grids (200 mesh; Quantifoil Micro Tools, GmbH) with or without a layer of continuous carbon (2 nm). Grids were glow discharged beforehand using a Gatan Solarus 950 (Gatan, Inc.) plasma cleaner for 30 s. The grids were blotted for 0.5 s at room temperature and 85% relative humidity and plunge frozen in liquid ethane using a Vitrobot Mark IV (Thermo Fisher Scientific). Screening of the grids showed better distribution of particles on grids with continuous carbon layer, which accordingly were selected for high-resolution data acquisition. Data were acquired in one session on a Titan Krios G1 (Thermo Fisher Scientific) operated at 300 kV, equipped with a K3 camera operated by Gatan Microscopy Suite (version 3.32, Gatan, Inc.) and a Quanta energy filter. Data were acquired in dose fractionation mode (30 frames) at a pixel size of 1.07 Å with a cumulative dose of 68.9 e⁻/Å² and a dose rate of 27.9 e⁻/px/s. Data were acquired in EPU (version 2.6, Thermo Fisher Scientific) using aberration free image shift (AFIS) with 4 images per hole and a nominal defocus range of −1 to −3 μm.

## Cryo-EM data processing

12,615 frame stacks were motion-corrected using MotionCor2[60] and CTF parameters were estimated using Gctf. Particle picking was performed in Warp[61] using the BoxNet[61] convolutional neural network. The model was trained in several consecutive iterations to eventually pick 1,060,446 particles that were extracted at a box size of 512 pixels at full spatial resolution (1.07 Å) and immediately binned to a box size of 128 pixels, corresponding to a pixel size of 4.28 Å. Particles were transferred to cryoSPARC[62] to run 2D classification, from which the best classes including 82,776 particles were selected. Selected particles were transferred to Relion 3.1 for re-centering and re-extraction at a box size of 256 pixels, corresponding to a pixel size of 2.14 Å. Particles were transferred back to cryoSPARC for ab-initio reconstruction. The resulting 3D reconstruction was used as a reference for subsequent 3D homogeneous refinement. The final reconstruction reached 7.7 Å global resolution.

## Neural network-based augmin model predictions and comparative model analysis

Models of the individual augmin TIII and TII tetramers, the TIII H3/H5-arm in complex with TII, as well as the augmin holocomplex were predicted in AlphaFold v2.2.0 (AlphaFold-Multimer) by submitting FASTA files with sequences of proteins contained in the respective complexes (augmin holocomplex: HAUS1-8; TIII: HAUS1,3,4,5; TII: HAUS2,6,7,8; TIII H3/H5-arm + TII: HAUS3(96-434), HAUS5(80-489), HAUS2(1-222), HAUS6(1-478), HAUS7(1-348), HAUS8(155-367)). AlphaFold-Multimer settings were as following: multimer preset, full databases, omitting relaxation step and setting maximum template date to 2050/01/01.

To distinguish differences in the predicted models, they were superposed using the matchmake command in UCSF Chimera[63]. For TIII, the full models were superposed. The TII models were superposed either according to the N-clamp (HAUS2: 1-117, HAUS6: 1-267, HAUS7: 1-

270, HAUS8: 155-260) or according to the C-clamp (HAUS2: 118-222, HAUS6: 268-398, HAUS7: 271-348, HAUS8: 261-367). For the TIII H3/H5-arm in complex with TII, all predicted models were superposed to the H3/H5-arm of the highest-scoring TIII model (HAUS3 198-242, 364-392 and HAUS5 172-275, 419-446). RMSD colouring for the predicted models was performed in PyMOL (PyMOL v2.1, Schrödinger) using the two most distinct conformations for each ensemble and applying colouring scale from 0 Å to 10 Å.

The predicted models of the TIII tetramer were rigid-body docked into the cryo-EM density of the TIII tetramer in UCSF Chimera[63] by simulating densities at 8 Å resolution and the resulting cross-correlation coefficient was measured for each model.

## Molecular dynamics flexible fitting of TIII into the cryo-EM density

The model with the highest prediction score was refined in Namdinator[64] using MDFF with 2000 minimisation steps and 20,000 simulation steps with 0 phenix real space refinement cycles.

## Generating models for the augmin holocomplex in two conformations

In step 1, the highest-scoring model predicted for the TIII H3/H5-arm in complex with TII was superposed to the highest-scoring model for TIII using the matchmake command in UCSF Chimera[63], according to the following residues: HAUS3 198-242, 364-392 and HAUS5 172-275, 419-446. The two models were merged in Coot[65] by retaining complementary segments of HAUS3 and HAUS5 from the two predicted models: (i) the model predicted for the H3/H5-arm in complex with TII contributed residues 198−364 from HAUS3 and residues 275-419 from HAUS5. The model predicted for TIII contributed the remaining segments of HAUS3 and HAUS5, as well as HAUS 1 and HAUS4.

In step 2, the resulting composite model of TIII in complex with the TII C-clamp was docked as a rigid body into the negative stain EM density of the augmin holocomplex in the open conformation using UCSF Chimera[63]. Afterwards, the TII N-clamp (HAUS2: 1-114, HAUS6: 1-263, HAUS7: 1-261, HAUS8: 155-259) was docked as a rigid body into the remaining density using UCSF Chimera[63]. Residues in the N-clamp hinge region (HAUS2: 115-119, HAUS6: 264-275, HAUS7:262-270, HAUS8: 260-269) and the residual disordered segments of HAUS6 (398-478) were deleted. Step 2 was repeated using the negative stain 3D reconstruction of the augmin holocomplex in the closed conformation.

## Structural analysis of the TII composite microtubule binding site

To analyse structural similarity of CH domains, the Ndc80 (PDB 3IZ0)[39] CH domain (residues 79-202) and EB3 (PDB 3JAR)[41] CH domain (residues 1−131) were superposed onto the predicted HAUS6 (residues 1-146) and HAUS7 CH domains (residues 1−125) in UCSF ChimeraX[66] and the RMSD between their protein backbones was measured using the match command.

To visualise binding of the Ndc80 and EB3 CH domains to microtubules, cryo-EM-derived models of Ndc80 bound to the α/β tubulin dimer (PDB 3IZ0) and EB3 bound to the microtubule lattice (PDB 3JAR) were superposed onto the microtubule lattice (PDB 6EW0)[67]. To extrapolate augmin binding to the microtubule the HAUS6 or HAUS7 CH domains were superposed onto Ndc80 CH domain.

To measure (relative) HAUS6 and HAUS7 CH domain rotations required to reach a configuration compatible with microtubule binding in tandem, as observed for the Ndc80 CH domains, the HAUS6 and HAUS7 CH domains were first centred on the Ndc80 CH domains bound to the microtubule lattice (PDB 3IZ0) according to CH domain centroids. The HAUS6 and HAUS7 CH domains were then separately superposed onto the Ndc80 CH domains using the matchmake command in UCSF Chimera and the overall rotation was obtained by the measure rotation command.

**Reporting summary**

Further information on research design is available in the Nature Research Reporting Summary linked to this article.

## Data availability

Atomic coordinates and the associated negative stain EM and cryo-EM densities have been deposited in the Protein Data Bank and the Electron Microscopy Data Bank under accession codes PDB 8AT2 /EMD-15631, PDB 8AT3 /EMD-15632, PDB 8AT4 /EMD-15633. Models predicted by AlphaFold-Multimer are available at the ModelArchive (https://modelarchive.org/) with the identifiers ma-wpr7k, ma-wsse3, ma-w3l0m, ma-8yvsa. The mass spectrometry proteomics data have been deposited in the ProteomeXchange Consortium[68] via the PRIDE[69] [http://www.proteomexchange.org/] partner repository with the dataset identifier PXD034895. Source data are provided with this paper. DNA constructs generated in this study are available upon request to the corresponding authors. Published structural data used in this article: PDB-3IZ0, PDB-3JAR, PDB-6EW0. Source data are provided with this paper.

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

## Acknowledgements

The authors thank Ursula Jäkle (ZMBH, Heidelberg) and Olga Kolesnikova (EMBL, Heidelberg) for support with protein expression and purification. We thank the Protein Expression and Purification Core Facility (EMBL, Heidelberg) for support and Alex Crowder (University of Calgary) for his help with Cross-linking MS data analysis. Moreover, the authors thank Oliver Gruss (University of Bonn) for discussion and Simone Reber (IRI, Humboldt University Berlin) providing cDNA constructs. We acknowledge the services SDS@hd and bwHPC supported by the Ministry of Science, Research and the Arts Baden-Württemberg, as well as the German Research Foundation (INST 35/1314-1 FUGG and INST 35/1134-1 FUGG). We also acknowledge access to the infrastructure of the Cryo-EM Network at the Heidelberg University (HDcryoNET) and support by Dirk Flemming (BZH, Heidelberg) and Götz Hofhaus (BioQuant, Heidelberg). This work is supported by grants of the Deutsche Forschungsgemeinschaft (DFG) to E.S. (DFG Schi 295/4-4) and to S.P. (DFG PF 963/1-4). S.P. also acknowledges funding by the Aventis Foundation and the 'Chica and Heinz Schaller' Foundation.

## Author contributions

M.W. performed cloning, protein expression, protein purification, and analysed negative stain electron microscopy data. A.B. contributed to biochemical characterization of protein complexes. E.Z. acquired and analysed cryo-EM data, analysed crosslinking mass spectrometry data, in silico experiments, created models, and visualised them. B.J.A.V. contributed to analysis and interpretation of structural data. A.N. performed negative staining and negative stain EM image acquisition. M.R. performed crosslinking mass spectrometry. T.H. performed AlphaFold-Multimer runs. S.E., E.S., and S.P. supervised the experiments. M.W., E.Z., E.S., and S.P. wrote the manuscript with input from the other authors. All authors discussed the data and gave final approval for publication.

## Funding

## Competing interests

The authors declare no competing interests.
