## [Peer Review File · Nature Communications]

REVIEWER COMMENTS

Reviewer #1 (Remarks to the Author):

In the manuscript “The augmin complex architecture reveals structural insights into microtubule branching” authors combined negative stain EM, cryo-EM, alpha-fold modelling and crosslinking mass spectrometry to generate an atomic model for *X. laevis* augmin complex. This structural model provides important insights into the molecular mechanism underlines microtubule branching.

In the integrative analysis, crosslinking mass spectrometry analysis was applied to validate the model of the TIII tetramer. The detected crosslinks showed good agreement with both the alpha fold generated model as well as the EM data. In addition, the clustered overlength crosslinks also confirmed the flexible regions in the tetramer. As the original mass spectrometry data was not provided for reviewing, the judgement could only be made based on the presented data in the manuscript. Overall, crosslinking mass spectrometry experiments were conducted properly. However additional details on data processing should be clarified before the manuscript being published in the Nature Communications.

1. The original crosslinking mass spectrometry data SHOULD be deposit in a public data depository (such as PRIDE) and made available to readers.
2. Authors stated that the identification of crosslinks was manually validated, please provide your criteria for manual validation.
3. Authors claimed that only crosslinks with <5% FDR were retained for further analysis, more details should be provided:
 - 1) are the FDR cut-off applied at peptide-spectrum match level, crosslinked peptide level or crosslinked residue pair level?
 - 2) are the FDR calculation performed separated for inter-protein and intra-protein crosslinks?
4. Authors only accept crosslinks linked to K on one end and KSTY on the other end. As BS3 crosslinks has symmetric reactivity, please explain why authors exclude crosslinks between residue S, T and Y (i.e. S-S, S-T, S-Y, T-T, T-Y and Y-Y) crosslinks.
5. Authors should also explain why they don't consider crosslinks to the N-termini of proteins.

Additional comments on the manuscript:

1. Crosslinking data was used often when discuss the structure of the augmin TIII tetramer, it would be easier for reader if crosslinks were also displayed in Fig 3, panel c, d and 3. Alternative, in Fig 2h, the subunits of TIII tetramer should be coloured in the same colour code as in figure 3.
2. Authors pointed out that the TII clamp were observed in either open or closed conformations, what are the proportion of each conformation?

3. Why don't authors apply crosslinking mass spectrometry analysis on the augmin octameric complex?

Reviewer #2 (Remarks to the Author):

By integrating modern structure prediction algorithms, intermediate resolution EM reconstructions and crosslinking MS, Zupa et al present an approximate, but convincing model of a flexible augmin octamer and a useful insight into the mechanism of microtubule branching.

Specific comments and suggested corrections (could not locate page numbers in the submitted manuscript - will assume the title page is no. 1):

Introduction

1. Could authors elaborate on the consequences of the lack of augmin in different systems? It is not clear how severe they are.

2. p. 2, bottom - I think the authors wanted to say:

"In human cells, augmin-mediated microtubule branching is not limited to mitosis and meiosis, but is also centrally involved in organising the non-centrosomal..." - "only" in the current wording changes the meaning.

3. Authors first introduce some structural information about augmin in the penultimate paragraph (p. 3) and then say in the ultimate paragraph that: "Our understanding (...) is strongly hampered by the lack of structural information on the augmin complex". Perhaps "by insufficient structural information" or alike would be more appropriate.

4. In the same paragraph (p. 3): "We identified a composite microtubule binding site with high structural and compositional similarity to (...) Ndc80." This sentence is not very clear. How much do we know about it being composite? Does the disordered N-terminus of HAUS8 contribute much to this binding? How much do we know? (Will come back to this question). And why "compositional similarity" to Ndc80? Because of the hammerhead being formed by two CH domains?

5. Thereafter: "nucleate a branched microtubule" rather than "the" in my opinion. Please, try to review indefinite and definite articles throughout the manuscript.

Results

6. Not sure why *X. laevis* system was used, perhaps would be good to explain.

7. Bottom of p. 5, "bringing their N- and C-terminal segments into spatial proximity (Fig. 3a)" - they do not look exactly proximal and it is not obvious that BS3 crosslinks could bridge that distance. Perhaps, would be useful to indicate distances in Fig. 3 and maybe other figures (if possible, without introducing too much visual clutter) - please, consider.

8. 3/4 of p. 8: "we generated models for the complete augmin octameric complex in two distinct conformations (Fig. 5c, Supplementary Fig. 7d)" - really difficult to follow how these two complexes are related. If authors think this is important, both conformations should be in the main manuscript and compared properly (label "open", "closed"). Would also suggest improving the presentation of the dynamics by potentially overlaying the fan-shaped ensemble of modelled structures with the two EM densities? Do these coincide well?

9. Closing sentences of the Results section (middle of p. 9) sound very speculative. How can one know about the "two-step" mechanism, etc. e.g. "...initial augmin recruitment to the microtubules through the flexible HAUS8 is followed by orientation..." - do we know anything about the sequence of these events? Would not harm to delete these sentences in my view, but that would impact on Fig. 6, too.

10. Alas, not sure about Fig. 6 - if authors feel strongly about proposing such sequence of events, perhaps that's all right, if clearly stated that this is purely speculative, but reducing it to one panel might also work, I suppose.

Discussion

11. Top of p. 10, should be: "...to address the structure of larger oligomers and to validate the predicted structures."

12. Middle of p. 10: "at a defined distance and in a defined orientation" - would agree about the distance, but orientation if probably only roughly defined at best, given the flexibility, which authors mention a couple of sentences later: "...the range of inter-microtubule angles...".

Methods

13. Maybe "DNA cloning" would be better?

14. Please, develop the abbreviation IDT

15. Please, provide the source of all types of EM grids

16. An error here?: "...6 classes and an offset search range of 20 pixels with an offset search range of 2 pixels."

17. Why did the authors use continuous carbon for cryo-EM? Would be useful to explain in the Methods.

Reviewer #3 (Remarks to the Author):

In this study the authors investigate the relationship between the structure of the Augmen complex and its ability to mediate microtubule branching through its ability to bind both the microtubule lattice and components of the γ -TuRC. Prediction of the structure of Augmen subcomplexes with validation using negative stain EM and single particle cryoEM as well as cross-linking MS provided new insight into how the elongated structure of the complex is critical to its function in microtubule branching. While the components involved in γ -TuRC and microtubule binding are distinct, they are functionally coupled by a flexible linker region. As well, the authors characterized the microtubule binding interface. This analysis is extremely detailed and provides a dynamic view of conformational states that contribute to functional plasticity of Augmen dependent microtubule branching to produce variations in microtubule- γ -TuRC angles that produce a variety of branched networks.

In the discussion, the authors state that the plasticity in angle of the γ -TuRC relative to that of the microtubule is consistent with the range of inter-microtubule angles observed in TIRF studies. This is a very brief statement and should be expanded with additional detail in the discussion. However, microtubule branching occurs in different cell types and the study would be improved by the additional comparison with in vivo angles for branched microtubules. If not already available, information is certainly accessible using super resolution microtubule methods that should be available to the authors.

The study would be considerably strengthened by an in vitro demonstration that validates the stated dependencies for the proposed two-step mechanism shown in figure 6.

The quality of the writing could be improved.

Point-by-Point Reply to Reviewers

We thank the reviewers for their thoughtful and very supportive comments. Below we address the specific points raised by the reviewers and elaborate on the corresponding changes in the manuscript.

Reviewer #1 (Remarks to the Author):

In the manuscript “The augmin complex architecture reveals structural insights into microtubule branching” authors combined negative stain EM, cryo-EM, alpha-fold modelling and crosslinking mass spectrometry to generate an atomic model for *X. laevis* augmin complex. This structural model provides important insights into the molecular mechanism underlines microtubule branching.

We thank the Reviewer for the positive evaluation of our manuscript.

In the integrative analysis, crosslinking mass spectrometry analysis was applied to validate the model of the TIII tetramer. The detected crosslinks showed good agreement with both the alpha fold generated model as well as the EM data. In addition, the clustered overlength crosslinks also confirmed the flexible regions in the tetramer. As the original mass spectrometry data was not provided for reviewing, the judgement could only be made based on the presented data in the manuscript.

Overall, crosslinking mass spectrometry experiments were conducted properly. However additional details on data processing should be clarified before the manuscript being published in the Nature Communications.

We have now adapted the manuscript per all requests and included the additional details in the methods section (p. 26).

1. The original crosslinking mass spectrometry data SHOULD be deposit in a public data depository (such as PRIDE) and made available to readers.

We have now submitted the original data to PRIDE with the dataset identifier PXD034895. The data will be publicly released prior to publication of the manuscript, but is already accessible using the following reviewer account:

Username: reviewer_pxd034895@ebi.ac.uk
Password: R9A3jmk

We have added the information to the Data availability section: “The mass spectrometry proteomics data have been deposited the ProteomeXchange Consortium (Deutsch et al., 2020) via the PRIDE (Perez-Riverol et al., 2022) [<http://www.proteomexchange.org/>] partner repository with the dataset identifier PXD034895.”

2. Authors stated that the identification of crosslinks was manually validated, please provide your criteria for manual validation.

Identified crosslinked peptides were manually validated according to MS spectra quality, MS/MS spectra quality, mass accuracy, peak assignment in MS/MS spectra and fragmentation of crosslink product (ions coverage of crosslinked peptides), as recommended in (Iacobucci and Sinz, 2017).

3. Authors claimed that only crosslinks with <5% FDR were retained for further analysis, more details should be provided:

1) are the FDR cut-off applied at peptide-spectrum match level, crosslinked peptide level or crosslinked residue pair level?

FDR estimation was controlled at the peptide pair and residue pair level in a feed-forward manner, as implemented in Mass Spec Studio (Sarpe et al., 2016) which we used for data processing.

2) are the FDR calculation performed separated for inter-protein and intra-protein crosslinks?

FDR was estimated separately for intra and inter-protein crosslinks.

4. Authors only accept crosslinks linked to K on one end and KSTY on the other end. As BS3 crosslinks has symmetric reactivity, please explain why authors exclude crosslinks between residue S, T and Y (i.e. S-S, S-T, S-Y, T-T, T-Y and Y-Y) crosslinks.

Crosslink residue pairs were constrained on K on one end and KSTY on the other end to reduce search space while not significantly affecting the crosslink distance.

5. Authors should also explain why they don't consider crosslinks to the N-termini of proteins.

N-termini of proteins were not considered for crosslinks, because this was not supported in our version of Mass Spec Studio.

Additional comments on the manuscript:

1. Crosslinking data was used often when discuss the structure of the augmin TIII tetramer, it would be easier for reader if crosslinks were also displayed in Fig 3, panel c, d and 3. Alternative, in Fig 2h, the subunits of TIII tetramer should be coloured in the same colour code as in figure 3.

We agree with the reviewer that the original visualization of crosslinks was not ideal for detailed visual inspection. We have therefore decided to include an additional figure (Supplementary Fig. 6), in which we show the crosslinks mapped back to the TIII tetramer model color-coded as in Fig. 3 and with zoomed views on clusters of crosslinks.

2. Authors pointed out that the TII clamp were observed in either open or closed conformations, what are the proportion of each conformation?

The relative proportions of the two conformations can be approximated based on the number of particles contributing to the two negative stain 3D classes, as given in the methods section. We now provide this information in the results section focusing on conformational plasticity of augmin (p. 9, top): "... EM densities representing the TII clamp in at least two different conformational states (Supplementary Fig. 9a), with a ratio of 1:1 as judged by number of particles contributing to the two individual classes (see methods)."

3. Why don't authors apply crosslinking mass spectrometry analysis on the augmin octameric complex?

We have conducted a crosslinking mass spectrometry experiment for the augmin octamer. However, the experiment was not conclusive, because we detected only very few high-confidence crosslinks for the modelled ordered part of the TII tetramer and no crosslinks for the TII-TIII interface, while we could reproduce crosslinks for the TIII. We therefore decided not to include these data into the manuscript.

Reviewer #2 (Remarks to the Author):

By integrating modern structure prediction algorithms, intermediate resolution EM reconstructions and crosslinking MS, Zupa et al present an approximate, but convincing model of a flexible augmin octamer and a useful insight into the mechanism of microtubule branching.

We thank the Reviewer for the positive evaluation of our manuscript.

Specific comments and suggested corrections (could not locate page numbers in the submitted manuscript - will assume the title page is no. 1):

Introduction

1. Could authors elaborate on the consequences of the lack of augmin in different systems? It is not clear how severe they are.

We have revised the text according to the Reviewer's suggestion and included a dedicated section in the introduction (p. 2, bottom):

"The essential role of augmin in microtubule branching was established across different species *in vivo*, including plant species (Hotta et al., 2012; Nakaoka et al., 2012), human cells (Funk et al., 2021), *Drosophila*, where depletion of augmin subunits results in spindle microtubule defects (Goshima et al., 2008; Meireles et al., 2009) and *X. laevis*, where augmin was shown to be central for spindle formation and integrity (Petry et al., 2011, 2013). In human cells, augmin-mediated microtubule branching is not limited to mitosis and meiosis, but it is also centrally involved in organising the non-centrosomal neuronal microtubule networks and in establishing axonal microtubule polarity (Sanchez-Huertas et al., 2016), thereby contributing to development of the central nervous system in mammals (Cunha-Ferreira et al., 2018; Viais et al., 2021). Consistently, knock down of the augmin subunit HAUS6 is lethal for mouse embryos (Watanabe et al., 2016)."

2. p. 2, bottom - I think the authors wanted to say: "In human cells, augmin-mediated microtubule branching is not limited to mitosis and meiosis, but is also centrally involved in organising the non-centrosomal..." - "only" in the current wording changes the meaning.

We have revised the sentence accordingly.

3. Authors first introduce some structural information about augmin in the penultimate paragraph (p. 3) and then say in the ultimate paragraph that: "Our understanding (...) is strongly hampered by the lack of structural information on the augmin complex". Perhaps "by insufficient structural information" or alike would be more appropriate.

We have revised the sentence accordingly.

4. In the same paragraph (p. 3): "We identified a composite microtubule binding site with high structural and compositional similarity to (...) Ndc80." This sentence is not very clear. How much do we know about it being composite? Does the disordered N-terminus of HAUS8 contribute much to this binding? How much do we know? (Will come back to this question).

Previous *in vitro* experiments have established differential contributions of TII subunits to microtubule binding affinity (Hsia et al., 2014; Song et al., 2018; Wu et al., 2008). Upon deletion of the disordered N-terminal segment of HAUS8, no microtubule binding could be observed anymore, indicating that the HAUS8 N-terminal segment is essential for microtubule binding of TII. However, the isolated HAUS8 N-terminal segment binds with 10-fold lower affinity to microtubules as compared to the full TII tetramer, indicating that additional contacts between TII and the microtubule lattice are important for stable augmin recruitment. In our study, we identify the HAUS6 and HAUS7 CH domains as the most likely candidates to mediate the additional interactions, rendering the N-clamp HH a composite microtubule binding module.

To clarify this, we have now included a more detailed description in the results (p. 9, middle) and discussion sections (p. 12, bottom), which also address comments 9 and 10 of Reviewer #2 and comment 4 of Reviewer #3.

And why "compositional similarity" to Ndc80? Because of the hammerhead being formed by two CH domains?

We consider the microtubule binding sites of augmin and the Ndc80 complex to be compositionally similar, because both binding sites are composed of a disordered and positively charged segment, accompanied by two CH domains.

We now explain in detail in the discussion section (p. 12, middle): "Not only the presence of two CH domains but the entire molecular organisation of the composite microtubule binding site in augmin is highly reminiscent of the kinetochore complex, where two CH domains contributed by Ndc80 and Nuf2 (corresponding to the HAUS6 and HAUS7 CH domains) and an unordered positively charged extension contributed by Ndc80 (corresponding to the HAUS8 N-terminus) are required for efficient microtubule-binding."

5. Thereafter: "nucleate a branched microtubule" rather than "the" in my opinion. Please, try to review indefinite and definite articles throughout the manuscript.

We have corrected the manuscript accordingly.

Results

6. Not sure why *X. laevis* system was used, perhaps would be good to explain.

We chose to work with the augmin complex from *X. laevis* for three reasons:

1) *X. laevis* is a popular model organism for studying processes related to microtubule nucleation, including microtubule branching, and a wide range of published data is available on these processes in *X. laevis*.

2) Among the data available already, it had been experimentally established that *X. laevis* augmin TII and TIII have differential binding behavior towards microtubules and the γ -TuRC, respectively, and that the tetramers form stable subcomplexes after recombinant expression, which we considered a clear advantage for experimental work, but also for computational structure prediction.

3) In previous work (Liu et al., 2019), we had established the purification of endogenous γ -TuRC from *X. laevis* egg extract and we wanted to maintain full compatibility between γ -TuRC and augmin for future lines of research.

We have modified the second paragraph in the results section where we now mention some of these aspects (p. 4, middle):

“We focused on the augmin TII and TIII tetramers from *X. laevis*, which had been experimentally reported to form stable subcomplexes on their own (Song et al., 2018) and are thought to represent two independent functional modules for microtubule and γ -TuRC binding, respectively (Song et al., 2018).”

7. Bottom of p. 5, "bringing their N- and C-terminal segments into spatial proximity (Fig. 3a)" - they do not look exactly proximal and it is not obvious that BS3 crosslinks could bridge that distance. Perhaps, would be useful to indicate distances in Fig. 3 and maybe other figures (if possible, without introducing too much visual clutter) - please, consider.

We fully agree that the very terminal segments of HAUS3 and HAUS5 are too distant to be crosslinked with BS3. We phrased this sentence more carefully now: "HAUS3 and HAUS5 fold back onto each other, bringing segments located more towards their protein termini into spatial proximity (Fig. 3a), ..."

Furthermore, in response to a comment of Reviewer #1, we have included a new figure (Supplementary Fig. 6) in which we show the crosslinks mapped back to the atomic model of the TIII tetramer in more detail.

8. 3/4 of p. 8: "we generated models for the complete augmin octameric complex in two distinct conformations (Fig. 5c, Supplementary Fig. 7d)" - really difficult to follow how these two complexes are related. If authors think this is important, both conformations should be in the main manuscript and compared properly (label "open", "closed").

In response, we have reorganized the manuscript and present the second augmin conformation in a dedicated paragraph in the main text, which should make it easier to follow (p. 8, top): “The N-clamp is linked to the TII C-clamp by a hinge region, which allows for an ‘opening and closing’ motion of the TII clamp, as suggested by the ensemble of predicted models (Figs. 1d, 5a). Consistently, negative stain 2D classes of the augmin holocomplex reflected a range of TII conformations (Fig. 4b, Supplementary Fig. 7e) and 3D classification provided EM densities representing the TII clamp in at least two different conformational states (Supplementary Fig. 9a), with a ratio of 1:1 as judged by number of particles contributing to the two individual classes (see methods). Following the same modelling approach as described above, we also generated a model for the second TII conformation (Supplementary Fig. 9b-d). The two experimentally observed TII conformations are related by an opening/closing of the TII clamp of approximately 23° (Fig. 5d).”

Furthermore, we included a comparison of the two conformations as a new panel in Fig. 5, in which the conformations are labeled ‘open’ and ‘closed’.

Would also suggest improving the presentation of the dynamics by potentially overlaying the fan-shaped ensemble of modelled structures with the two EM densities? Do these coincide well?

We thank the reviewer for this suggestion. In fact, during initial analysis, we observed that in none of the predicted models, the TII N-clamp orientation was compatible with the octamer EM densities, indicating that AlphaFold-Multimer was not capable of correctly predicting the relative arrangement of the TII N- and C-clamp around the hinge and highlighting the need for orthogonal structural data to validate the predicted models. To generate models for the augmin holocomplex, we thus removed the wrongly predicted hinge region and docked the TII N-clamp as a rigid body into the remaining density segments, guided by the orientation of the N-clamp HH. We now emphasize this aspect more clearly in the results (p. 8, bottom) and discussion (p. 10, middle) sections of the manuscript and also included a new figure (Supplementary Fig. 8) to visualize the mismatch between the ensemble of predicted models and the EM densities in detail.

9. Closing sentences of the Results section (middle of p. 9) sound very speculative. How can one know about the "two-step" mechanism, etc. e.g. "...initial augmin recruitment to the microtubules through the flexible HAUS8 is followed by orientation..." - do we know anything about the sequence of these events? Would not harm to delete these sentences in my view, but that would impact on Fig. 6, too.

Please refer to our reply to comment 3 of Reviewer #3.

10. Alas, not sure about Fig. 6 - if authors feel strongly about proposing such sequence of events, perhaps that's all right, if clearly stated that this is purely speculative, but reducing it to one panel might also work, I suppose.

Please refer to our reply to comment 3 of Reviewer #3.

Discussion

11. Top of p. 10, should be: "...to address the structure of larger oligomers and to validate the predicted structures."

We have corrected the manuscript accordingly.

12. Middle of p. 10: "at a defined distance and in a defined orientation" - would agree about the distance, but orientation if probably only roughly defined at best, given the flexibility, which authors mention a couple of sentences later: "...the range of inter-microtubule angles...".

We agree with the reviewer and have revised the sentence accordingly: "at a defined distance and an approximate orientation with respect to the pre-existing microtubule."

Methods

13. Maybe "DNA cloning" would be better?

We have changed the manuscript accordingly.

14. Please, develop the abbreviation IDT

We have corrected the manuscript accordingly.

15. Please, provide the source of all types of EM grids

We now provide sources of all EM grid types.

16. An error here?: "...6 classes and an offset search range of 20 pixels with an offset search range of 2 pixels."

We have corrected the manuscript accordingly: "...6 classes and an offset search range of 20 pixels with an offset search step of 2 pixels."

17. Why did the authors use continuous carbon for cryo-EM? Would be useful to explain in the Methods.

During screening of grids, we observed a better particle distribution on grids covered with a layer of continuous carbon. We included an explanation in the method section: "Screening of the grids showed better distribution of particles on grids with continuous carbon layer, which accordingly were selected for high-resolution data acquisition."

Reviewer #3 (Remarks to the Author):

In this study the authors investigate the relationship between the structure of the Augmen complex and its ability to mediate microtubule branching through its ability to bind both the microtubule lattice and components of the γ -TuRC. Prediction of the structure of Augmen subcomplexes with validation using negative stain EM and single particle cryoEM as well as cross-linking MS provided new insight into how the elongated structure of the complex is critical to its function in microtubule branching. While the components involved in γ -TuRC and microtubule binding are distinct, they are functionally coupled by a flexible linker region. As well, the authors characterized the microtubule binding interface. This analysis is extremely detailed and provides a dynamic view of conformational states that contribute to functional plasticity of Augmen dependent microtubule branching to produce variations in microtubule- γ -TuRC angles that produce a variety of branched networks.

In the discussion, the authors state that the plasticity in angle of the γ -TuRC relative to that of the microtubule is consistent with the range of inter-microtubule angles observed in TIRF studies. This is a very brief statement and should be expanded with additional detail in the discussion. However, microtubule branching occurs in different cell types and the study would be improved by the additional comparison with *in vivo* angles for branched microtubules. If not already available, information is certainly accessible using super resolution microtubule methods that should be available to the authors.

We thank the reviewer for this comment and fully agree that it would be of great interest to define individual steps and dynamics of augmin recruitment and microtubule branching geometry in detail, for instance using super resolution light microscopy methods.

However, we believe that such challenging and time-consuming analyses are going beyond the scope of our current study, which focuses on elucidating the structure of the augmin complex using integrative structural biology. We therefore rather prefer to discuss on the basis of already published data how conformational plasticity of TII may relate to the range of microtubule branching angles observed using TIRF microscopy both *in vitro* and *in vivo* (p. 11, top):

“The TII hinge region separating the two clamp elements may allow for a defined range of flexibility in the relative positioning of the γ -TuRC with respect to the pre-existing microtubule. Using TIRF microscopy to analyze the geometry of branched microtubule networks *in vitro* ((Alfaro-Aco et al., 2020) and (Zhang et al., 2022)) and also *in vivo* (Verma and Maresca, 2019) it has been established that the inter-microtubule angles after branching have a spread of approximately 20-30° around the mean branching angle. This is in good agreement with the opening angle experimentally observed (23°; Fig. 5d) for the TII clamp, suggesting that plasticity of the TII hinge region may directly define the variability of microtubule branching angles.”.

The study would be considerably strengthened by an *in vitro* demonstration that validates the stated dependencies for the proposed two-step mechanism shown in figure 6.

While we agree that in-depth analysis of the series of events during augmin recruitment to microtubules using tailored and structure-guided *in vitro* experiments would be ideal, again we believe that this would be a very challenging and time-consuming study going beyond the scope of this manuscript, in particular because published data already support key aspects of the model we propose.

Published data (Hsia et al., 2014) demonstrated that TII binding to microtubules is entirely dependent on the presence of the HAUS8 N-terminus, while binding affinity of the TII tetramer to microtubules is 10 times higher as compared to the isolated HAUS8 N-terminus, showing that additional TII components are required for stable binding. Integrating these data with our new structural data indicates a role of the flexible HAUS8 N-terminus in the initial stages of microtubule binding, most likely in establishing the initial interactions between augmin and the microtubule, followed by stabilization of the interaction and positioning of augmin on the microtubule by HAUS6 / HAUS7 CH domain binding. Furthermore, such a binding process involving two different modes of interaction would be consistent with the dynamics of augmin binding to microtubules as very recently characterized *in vitro* using TIRF microscopy (Zhang et al., 2022). In this study, augmin was observed to diffuse along the microtubule before it stably binds and the branching reaction occurs.

In response to this comment, as well as comments #9 & #10 from Reviewer #2, we now relate our model in more detail with the published interaction data (Hsia et al., 2014; Wu et al., 2008) and we now more clearly point out that it is still a speculative model by:

- 1) ...presenting the model at the end of the discussion section (instead of the results section).
- 2) ... carefully phrasing the text.
- 3) ...integrating the relevant pieces of published data (including the citations) into the figure legend of the model figure.

Finally, as suggested by Reviewer #2, we re-organized our model figure. We removed step 3 from the figure and show HAUS8 N-terminus binding and CH domain binding in one panel, to reduce the impression of these events being strictly sequential.

The updated discussion section (p. 12, bottom) now reads: “Finally, we integrated our structural data with available interaction data to propose a hypothetical model for augmin recruitment to microtubules (Figure 6). TII binding to microtubules was demonstrated to be fully dependent on the presence of the HAUS8 N-terminus, although the HAUS8 N-terminus is binding to microtubules with comparably low affinity (Hsia et al., 2014), which together may suggest a role of the highly flexible HAUS8 N-terminus in establishing the initial interactions between augmin and the microtubule. Binding affinity of the TII tetramer to microtubules is 10 times higher as compared to the isolated HAUS8 N-terminus(Hsia et al., 2014), which clearly indicates that additional TII components are required for stable binding. These were identified

in our study to most likely correspond to the CH domains of HAUS6 and HAUS7, located in the N-clamp HH. Thus, initial augmin binding via the HAUS8 N-terminus may potentially be followed by stabilisation of the interaction and positioning of augmin on the microtubule for the branching reaction by HAUS6/7 CH domain binding. Such a binding process involving two different modes of interaction would be consistent with the dynamics of augmin binding to microtubules as recently observed *in vitro* using TIRF microscopy, where augmin has the capability to diffuse along the microtubule before it stably binds and the branching reaction occurs (Zhang et al., 2022).”

The quality of the writing could be improved.

We carefully revised the text of the entire manuscript.

References

- Alfaro-Aco, R., Thawani, A., and Petry, S. (2020). Biochemical reconstitution of branching microtubule nucleation. *Elife* 9.
- Cunha-Ferreira, I., Chazeau, A., Buijs, R.R., Stucchi, R., Will, L., Pan, X., Adolfs, Y., van der Meer, C., Wolthuis, J.C., Kahn, O.I., et al. (2018). The HAUS Complex Is a Key Regulator of Non-centrosomal Microtubule Organization during Neuronal Development. *Cell Rep.* 24, 791–800.
- Deutsch, E.W., Bandeira, N., Sharma, V., Perez-Riverol, Y., Carver, J.J., Kundu, D.J., García-Seisdedos, D., Jarnuczak, A.F., Hewapathirana, S., Pullman, B.S., et al. (2020). The ProteomeXchange consortium in 2020: enabling ‘big data’ approaches in proteomics. *Nucleic Acids Res.* 48, D1145–D1152.
- Funk, L., Su, K.-C., Feldman, D., Singh, A., Moodie, B., Blainey, P.C., and Cheeseman, I.M. (2021). The phenotypic landscape of essential human genes. *BioRxiv* 2021.11.28.470116.
- Goshima, G., Mayer, M., Zhang, N., Stuurman, N., and Vale, R.D. (2008). Augmin: a protein complex required for centrosome-independent microtubule generation within the spindle. *J. Cell Biol.* 181, 421–429.
- Hotta, T., Kong, Z., Ho, C.-M.K., Zeng, C.J.T., Horio, T., Fong, S., Vuong, T., Lee, Y.-R.J., and Liu, B. (2012). Characterization of the Arabidopsis Augmin Complex Uncovers Its Critical Function in the Assembly of the Acentrosomal Spindle and Phragmoplast Microtubule Arrays. *Plant Cell* 24, 1494–1509.
- Hsia, K.-C., Wilson-Kubalek, E.M., Dottore, A., Hao, Q., Tsai, K.-L., Forth, S., Shimamoto, Y., Milligan, R.A., and Kapoor, T.M. (2014). Reconstitution of the augmin complex provides insights into its architecture and function. *Nat. Cell Biol.* 16, 852–863.
- Iacobucci, C., and Sinz, A. (2017). To Be or Not to Be? Five Guidelines to Avoid Misassignments in Cross-Linking/Mass Spectrometry. *Anal. Chem.* 89, 7832–7835.
- Liu, P., Zupa, E., Neuner, A., Böhrer, A., Loerke, J., Flemming, D., Ruppert, T., Rudack, T., Peter, C., Spahn, C., et al. (2019). Insights into the assembly and activation of the microtubule nucleator γ -TuRC. *Nature*.
- Meireles, A.M., Fisher, K.H., Colombié, N., Wakefield, J.G., and Ohkura, H. (2009). Wac: a new Augmin subunit required for chromosome alignment but not for acentrosomal microtubule assembly in female meiosis. *J. Cell Biol.* 184, 777–784.
- Nakaoka, Y., Miki, T., Fujioka, R., Uehara, R., Tomioka, A., Obuse, C., Kubo, M., Hiwatashi, Y., and Goshima, G. (2012). An Inducible RNA Interference System in *Physcomitrella patens* Reveals a Dominant Role of Augmin in Phragmoplast Microtubule Generation. *Plant Cell* 24, 1478–1493.
- Perez-Riverol, Y., Bai, J., Bandla, C., García-Seisdedos, D., Hewapathirana, S., Kamatchinathan, S., Kundu,

- D.J., Prakash, A., Frericks-Zipper, A., Eisenacher, M., et al. (2022). The PRIDE database resources in 2022: a hub for mass spectrometry-based proteomics evidences. *Nucleic Acids Res.* *50*, D543–D552.
- Petry, S., Pugieux, C., Nédélec, F.J., and Vale, R.D. (2011). Augmin promotes meiotic spindle formation and bipolarity in *Xenopus* egg extracts. *Proc. Natl. Acad. Sci. U. S. A.* *108*, 14473–14478.
- Petry, S., Groen, A.C., Ishihara, K., Mitchison, T.J., and Vale, R.D. (2013). Branching Microtubule Nucleation in *Xenopus* Egg Extracts Mediated by Augmin and TPX2. *Cell* *152*, 768–777.
- Sanchez-Huertas, C., Freixo, F., Viais, R., Lacasa, C., Soriano, E., and Luders, J. (2016). Non-centrosomal nucleation mediated by augmin organizes microtubules in post-mitotic neurons and controls axonal microtubule polarity. *Nat. Commun.* *7*, 12187.
- Sarpe, V., Rafiei, A., Hepburn, M., Ostan, N., Schryvers, A.B., and Schriemer, D.C. (2016). High Sensitivity Crosslink Detection Coupled With Integrative Structure Modeling in the Mass Spec Studio. *Mol. Cell. Proteomics* *15*, 3071–3080.
- Song, J.-G., King, M.R., Zhang, R., Kadzik, R.S., Thawani, A., and Petry, S. (2018). Mechanism of how augmin directly targets the gamma-tubulin ring complex to microtubules. *J. Cell Biol.* *217*, 2417–2428.
- Verma, V., and Maresca, T.J. (2019). Direct observation of branching MT nucleation in living animal cells. *J. Cell Biol.* *218*, 2829–2840.
- Viais, R., Fariña-Mosquera, M., Villamor-Payà, M., Watanabe, S., Palenzuela, L., Lacasa, C., and Lüders, J. (2021). Augmin deficiency in neural stem cells causes p53-dependent apoptosis and aborts brain development. *Elife* *10*, e67989.
- Watanabe, S., Shioi, G., Furuta, Y., and Goshima, G. (2016). Intra-spindle Microtubule Assembly Regulates Clustering of Microtubule-Organizing Centers during Early Mouse Development. *Cell Rep.* *15*, 54–60.
- Wu, G., Lin, Y.-T., Wei, R., Chen, Y., Shan, Z., and Lee, W.-H. (2008). Hice1, a novel microtubule-associated protein required for maintenance of spindle integrity and chromosomal stability in human cells. *Mol. Cell. Biol.* *28*, 3652–3662.
- Zhang, Y., Hong, X., Hua, S., and Jiang, K. (2022). Reconstitution and mechanistic dissection of the human microtubule branching machinery. *J. Cell Biol.* *221*, e202109053.

REVIEWERS' COMMENTS

Reviewer #1 (Remarks to the Author):

In the revised manuscript authors have made modifications according to the comments and improved the manuscript. Additional information has been provided in the rebuttal letter which has answered/addressed many questions from the reviewers.

However, there are still confusion remaining on the error estimation of the crosslinking datasets. A clear description on data process procedure would help readers to judge the accuracy and reliability of the crosslinking dataset. Authors should clarify following questions before sharing the crosslinking/MS data in this manuscript as a publication.

1. The authors make a 5% FDR cutoff based on E-score from Mass Spec Studio. What is the E-score applied as the cutoff (for example E-score of 10 or 5).
2. Was the score provided in the supplemental dataset 1,2 and 3 E-score? If it was a different score, the E-score should be also provided in the supplemental datasets.
3. Was the data listed in supplemental dataset 1 already after 5% FDR cutoff?
4. Authors also carried out manual validation on the dataset. Were all crosslinks or only the 190 crosslinks with score higher than 250 manually validated? There are 23331 linked residue pairs reported in Supplemental dataset 1.
5. How many decoy matches remain after manual validation and 250 E score cut-off. This number may provide a clue on the error rate of the data actually used for model validation.
6. Please provide references for how the separate inter- and intra- protein FDR estimation was conducted in the used software. As neither in the reference for the software nor in the source site of the software this information is provided.

Additional points:

1. crosslinking/MS is still a relatively new technique in comparison to EM or X ray crystallography. It would be good if authors provide a reference for the technique, such as a review.
2. Authors should add fragmentation methods for MS acquisition, as this information may be required by data processing software, for example Mass Spec Studio which was used in this study.

Reviewer #2 (Remarks to the Author):

Corrections are excellent and I have no further concerns about this manuscript.

Szymon W. Manka

Point-by-Point Reply to Reviewers

We thank the reviewers for their helpful comments on our revised manuscript, in particular regarding the suggested improvements for reporting of our crosslinking data. Below we address the specific points raised by the reviewers and elaborate on the corresponding changes in the manuscript.

Reviewer #1 (Remarks to the Author):

In the revised manuscript authors have made modifications according to the comments and improved the manuscript. Additional information has been provided in the rebuttal letter which has answered/addressed many questions from the reviewers.

However, there are still confusion remaining on the error estimation of the crosslinking datasets. A clear description on data process procedure would help readers to judge the accuracy and reliability of the crosslinking dataset. Authors should clarify following questions before sharing the crosslinking/MS data in this manuscript as a publication.

We have revised our description in the manuscript and answered the raised questions in detail. Some of the latest analysis features of the current Mass Spec Studio release are indeed not yet documented in a peer-reviewed article or pre-print. Yet, to improve the reporting in our manuscript we have been in direct contact with the developers of Mass Spec Studio and include them in the acknowledgements.

1. The authors make a 5% FDR cutoff based on E-score from Mass Spec Studio. What is the E-score applied as the cutoff (for example E-score of 10 or 5).

We apologize for not having described the workflow during filtering of spectra unambiguously. In the first processing step, we reduce database size by using only 70% of spectra based on an initial E-score, independent from FDR. In the second processing step using the reduced and refined database, we used standard FDR estimation based on the q-value and rejected spectra for which the q-value is higher than the expected rate of false discovery. Of note, in between these two steps, we manually validated all peptide spectrum matches from all mass spec acquisition runs (see point 4).

We have now described this more clearly in the updated methods section.

2. Was the score provided in the supplemental dataset 1,2 and 3 E-score? If it was a different score, the E-score should be also provided in the supplemental datasets.

In Supplemental Dataset files, we provide aggregate scores that Mass Spec Studio calculates based on the sum of spectrum scores that makeup the aggregate item. Spectrum scores are OMSSA (Geer et al., 2004)-based and reported as the $-\ln(\text{Expectation})$. The order of aggregation-levels that determine the aggregate scores is: spectrum-matches \rightarrow peptide-pairs \rightarrow residue-pairs \rightarrow protein-protein interactions, where each level is the sum of scores calculated at the previous level. This approach is very similar to XiFDR (Fischer and Rappsilber, 2017, 2018) with the exception that Mass Spec Studio uses the sum/L1-norm, rather than the Euclidean/L2-norm.

We have explained this now in more detail in the methods section and changed the 'score' label in Supplemental Data files to 'aggregate score'.

3. Was the data listed in supplemental dataset 1 already after 5% FDR cutoff?

Yes, the data listed in Supplemental Dataset 1 is after applying a cut-off at 5% FDR and manual validation of peptide spectrum matches (see point 4). We now specify this more clearly in the description of files.

4. Authors also carried out manual validation on the dataset. Were all crosslinks or only the 190 crosslinks with score higher than 250 manually validated? There are 23331 linked residue pairs reported in Supplemental dataset 1.

We manually validated all peptide spectrum matches from all mass spec acquisition runs after processing of the raw data and before exporting the list of cross-links from Mass Spec Studio.

We now state this more clearly in the updated methods section.

5. How many decoy matches remain after manual validation and 250 E score cut-off. This number may provide a clue on the error rate of the data actually used for model validation.

After manual validation and applying a cut-off at 250 aggregate score, no decoy matches remain.

6. Please provide references for how the separate inter- and intra-protein FDR estimation was conducted in the used software. As neither in the reference for the software nor in the source site of the software this information is provided.

There are no differences in data processing for intra- and intermolecular crosslinks, but the two types of crosslinks are separated right before the FDR estimation. FDR estimation for the separated input data was standard. We now clarified this in the updated methods section.

There is no specific reference yet for how Mass Spec Studio performs FDR estimation separately for inter- and intra-molecular crosslinks.

Additional points:

1. crosslinking/MS is still a relatively new technique in comparison to EM or X ray crystallography. It would be good if authors provide a reference for the technique, such as a review.

As suggested we added two references for reviews on crosslinking mass spectrometry in the results section, page 5:

“Subsequently, we sought to further validate our model of the TIII tetramer by crosslinking mass spectrometry (Graziadei and Rappsilber, 2022; O’Reilly and Rappsilber, 2018).”

2. Authors should add fragmentation methods for MS acquisition, as this information may be required by data processing software, for example Mass Spec Studio which was used in this study.

We used ‘higher energy collisional dissociation’ (HCD) fragmentation for MS acquisition.

We added this to the updated methods section of the manuscript.

Reviewer #2 (Remarks to the Author):

Corrections are excellent and I have no further concerns about this manuscript.

We thank the Reviewer for the positive evaluation of our revised manuscript.

References

Fischer, L., and Rappsilber, J. (2017). Quirks of Error Estimation in Cross-Linking/Mass Spectrometry. *Anal. Chem.* 89, 3829–3833.

Fischer, L., and Rappsilber, J. (2018). False discovery rate estimation and heterobifunctional cross-linkers. *PLoS One* 13, e0196672.

Geer, L.Y., Markey, S.P., Kowalak, J.A., Wagner, L., Xu, M., Maynard, D.M., Yang, X., Shi, W., and Bryant, S.H. (2004). Open mass spectrometry search algorithm. *J. Proteome Res.* 3, 958–964.

Graziadei, A., and Rappsilber, J. (2022). Leveraging crosslinking mass spectrometry in structural and cell biology. *Structure* 30, 37–54.

O'Reilly, F.J., and Rappsilber, J. (2018). Cross-linking mass spectrometry: methods and applications in structural, molecular and systems biology. *Nat. Struct. Mol. Biol.* 25, 1000–1008.